# A prospective observational study of post-COVID-19 chronic fatigue syndrome following the first pandemic wave in Germany and biomarkers associated with symptom severity

A subset of patients has long-lasting symptoms after mild to moderate Coronavirus disease 2019 (COVID-19). In a prospective observational cohort study, we analyze clinical and laboratory parameters in 42 post-COVID-19 syndrome patients (29 female/13 male, median age 36.5 years) with persistent moderate to severe fatigue and exertion intolerance six months following COVID-19. Further we evaluate an age- and sex-matched postinfectious non-COVID-19 myalgic encephalomyelitis/chronic fatigue syndrome cohort comparatively. Most post-COVID-19 syndrome patients are moderately to severely impaired in daily live. 19 post-COVID-19 syndrome patients fulfill the 2003 Canadian Consensus Criteria for myalgic encephalomyelitis/chronic fatigue syndrome. Disease severity and symptom burden is similar in post-COVID-19 syndrome/ myalgic encephalomyelitis/chronic fatigue syndrome and non-COVID-19/ myalgic encephalomyelitis/chronic fatigue syndrome patients. Hand grip strength is diminished in most patients compared to normal values in healthy. Association of hand grip strength with hemoglobin, interleukin 8 and C-reactive protein in post-COVID-19 syndrome/non-myalgic encephalomyelitis/chronic fatigue syndrome and with hemoglobin, N-terminal prohormone of brain natriuretic peptide, bilirubin, and ferritin in post-COVID-19 syndrome/ myalgic encephalomyelitis/chronic fatigue syndrome may indicate low level inflammation and hypoperfusion as potential pathomechanisms.

Infection with severe acute respiratory syndrome coronavirus type 2 (SARS-CoV-2) poses a major threat for developing chronic morbidity. While older patients or those with risk factors have a high possibility of severe or critical coronavirus disease 2019 (COVID-19), about 80% of COVID-19 cases are mild according to World Health Organization (WHO) criteria[1]. Soon there were reports, however, of patients with persistent symptoms following mild COVID-19 referred to as post-COVID syndrome (PCS) or long COVID[2,3]. Frequent symptoms include fatigue, impaired physical and cognitive function, headache, breathlessness, palpitations and many other symptoms impairing activities of daily living in many patients[4–9]. A patient survey of long COVID in young patients listed fatigue, post-exertional malaise (PEM), and cognitive dysfunction among the most frequent symptoms requiring reduced working hours in almost half and inability to work in 22% of patients[4]. PEM describes an intolerance to mental and physical exertion, which triggers an

✉ e-mail: claudia.kedor@charite.de

aggravation of symptoms typically lasting for more than 14 h up to several days[10].

Long-term health consequences following mild COVID-19 are poorly understood yet but have been feared based on observations from SARS-CoV-1. Here many patients were reported who developed a severe postinfectious syndrome with persistent fatigue, muscle pain, shortness of breath, and mental symptoms independent of illness severity[11]. Various pathogens including Epstein-Barr virus (EBV), enteroviruses, and dengue viruses are known to trigger myalgic encephalomyelitis/chronic fatigue syndrome (ME/CFS) in a subset of patients[12]. It is unclear yet if pathomechanisms of postinfectious fatigue syndromes may vary depending on the pathogen.

ME/CFS is a debilitating chronic disease with a worldwide prevalence of 0.3–0.8%[13]. Profound mental and physical fatigue, PEM, cognitive impairment, chronic pain, and orthostatic intolerance are key symptoms of ME/CFS. The best discriminating symptoms distinguishing ME/CFS from chronic fatigue in multiple sclerosis were flu-like symptoms and the intolerance to mental and physical exertion triggering PEM for more than 14 h[14]. ME/CFS is classified by the WHO as a neurological disease with G93.3 in the International Classification of Diseases 10th revision (ICD-10). Although the pathomechanisms are not well understood yet, there is ample evidence of immune, autonomous nervous system and metabolic dysregulation[15]. There is emerging evidence that postinfectious ME/CFS has an autoimmune mechanism[15–17].

We report here on the first results of our ongoing prospective observational cohort study initiated at Charité – Universitätsmedizin Berlin in August 2020 to characterize patients with persistent fatigue and exertion intolerance following mild to moderate COVID-19 according to WHO criteria and to assess whether they meet diagnostic criteria for ME/CFS[1,18]. Our study is a substudy of the Pa-COVID-19 study, a prospective observational cohort study assessing pathophysiology and clinical characteristics of patients with COVID-19 at Charité Universitätsmedizin Berlin[19]. This detailed description includes clinical characteristics and biomarker findings of patients from the first pandemic wave suffering from post-COVID-19 syndrome (PCS) at six months following COVID-19 diagnosis. Patients had COVID-19 between March and June 2020 when there were no variants of SARS-CoV-2 reported in our region. Due to the complexity of symptoms, patients were comprehensively evaluated by a team of medical professionals from various disciplines including clinical immunology, rheumatology, neurology, cardiology, endocrinology, and pulmonology with long-standing experience in diagnosing ME/CFS (https://cfc.charite.de). We hypothesized that COVID-19 can lead to a persistent fatigue syndrome which fulfills diagnostic criteria of ME/CFS and that patients suffering from ME/CFS display specific characteristics. Our findings confirm initial concerns that COVID-19 leads to persistent fatigue syndromes in a subset of young individuals following mild to moderate infectious disease.

## Results

### Demographic and baseline clinical characteristics

We report on a total of 42 patients who presented to the Charité Fatigue Center with PCS all suffering from persistent moderate to severe fatigue and exertion intolerance six months after diagnosis of COVID-19. Details of patient selection is reported in the methods section. Table 1 summarizes demographic and clinical characteristics of the study population. Most patients had mild COVID-19 ($n = 32$) and ten had moderate COVID-19 due to pneumonia, according to WHO criteria[1]. Three of the ten patients with pneumonia were treated in hospital but none of them required oxygen or mechanical ventilation thus making critical illness myopathy unlikely to explain any symptoms. Supplementary Table S1 shows the ten most frequent initial symptoms of COVID-19 reported by the patients.

19 of 42 PCS patients fulfilled the Canadian Consensus Criteria (CCC) for ME/CFS[18]. These patients are referred to as post-COVID-19 syndrome ME/CFS (PCS/ME/CFS), the other 23 patients are referred to as PCS/non-ME/CFS. Most PCS/non-ME/CFS patients (18 of 23) who did not fulfill ME/CFS criteria had exertion intolerance with a duration of PEM of <14 h. Furthermore, eight of 23 patients did not fulfill the CCC criteria for neurological/cognitive symptoms. Based on answers to the Patient Health Questionnaire 9 (PHQ9)[20], we have no evidence of severe depression in our study cohort. In two patients with an Epworth Sleepiness Scale (ESS) score[21] of >16 sleep apnea was excluded.

From all postinfectious non-COVID-19 ME/CFS patients evaluated during the same period at our clinic ($n = 123$) a sex- and age-matched control cohort who had the shortest duration of illness (13 months, range 7–19 months, $n = 19$) was selected. Their infectious trigger at disease onset is shown in Table 1.

The majority of patients from both the PCS and non-COVID ME/CFS cohort was severely impaired in daily life with a median Bell

## Table 1 | Demographic and baseline clinical characteristics

|  | PCS/non-ME/CFS ($n = 23$) | | PCS/ME/CFS ($n = 19$) | | non-COVID ME/CFS ($n = 19$) | |
|---|---|---|---|---|---|---|
|  | **Median** | **Range** | **Median** | **Range** | **Median** | **Range** |
| Age | 36 | (22–57) | 41 | (24–62) | 42 | (26–62) |
| Sex (female/male) | 15/8 |  | 14/5 |  | 14/5 |  |
| BMI | 22.5 | (18.0–36.2) | 24.3 | (18.1–31.8) | 21.3 | (18.5–27.6) |
| PHQ9 | 11 | (2–18) | 12 | (3–19) | nd | nd |
| ESS | 9 | (1–17) | 9 | (1–21) | nd | nd |
| Infectious trigger: |  |  |  |  |  |  |
| COVID-19 | 23 |  | 19 |  |  |  |
| Respiratory tract, unspecified |  |  |  |  | 9 |  |
| EBV |  |  |  |  | 3 |  |
| Gastroenteritis |  |  |  |  | 1 |  |
| Other infection |  |  |  |  | 6 |  |

Median and range of age, sex and body mass index (kg/m2) (BMI) are shown for PCS cohorts and non-COVID-ME/CFS. *PHQ9* Patient Health Questionnaire 9 (score 0–27), *ESS* Epworth Sleepiness Scale (score 0–24) and type of infection which triggered disease onset were assessed. *EBV* Epstein-Barr Virus; *nd* not determined. Data were analyzed using nonparametric all-pairs Dunn-type multiple contrast tests (age, sex, BMI) and Brunner-Munzel tests (PHQ9, ESS). Following BH correction all *p* values are = 1.

Patients enrolled in this study presented at our outpatient clinics 6 month after COVID-19 between August 2020 and November 2020. A sex- and age-matched control cohort of postinfectious non-COVID-19 ME/CFS patients evaluated during the same period at our clinic ($n = 123$) with the shortest duration of illness (13 months, range 7–19 months, $n = 19$) was selected. Source data are provided as a Source Data file.

*PCS/non-ME/CFS* post-COVID-19 syndrome/non-myalgic encephalomyelitis/chronic fatigue syndrome, *PCS/ME/CFS* post-COVID-19 syndrome/myalgic encephalomyelitis/chronic fatigue syndrome, *non-COVID ME/CFS* non-COVID-19 myalgic encephalomyelitis/chronic fatigue syndrome.

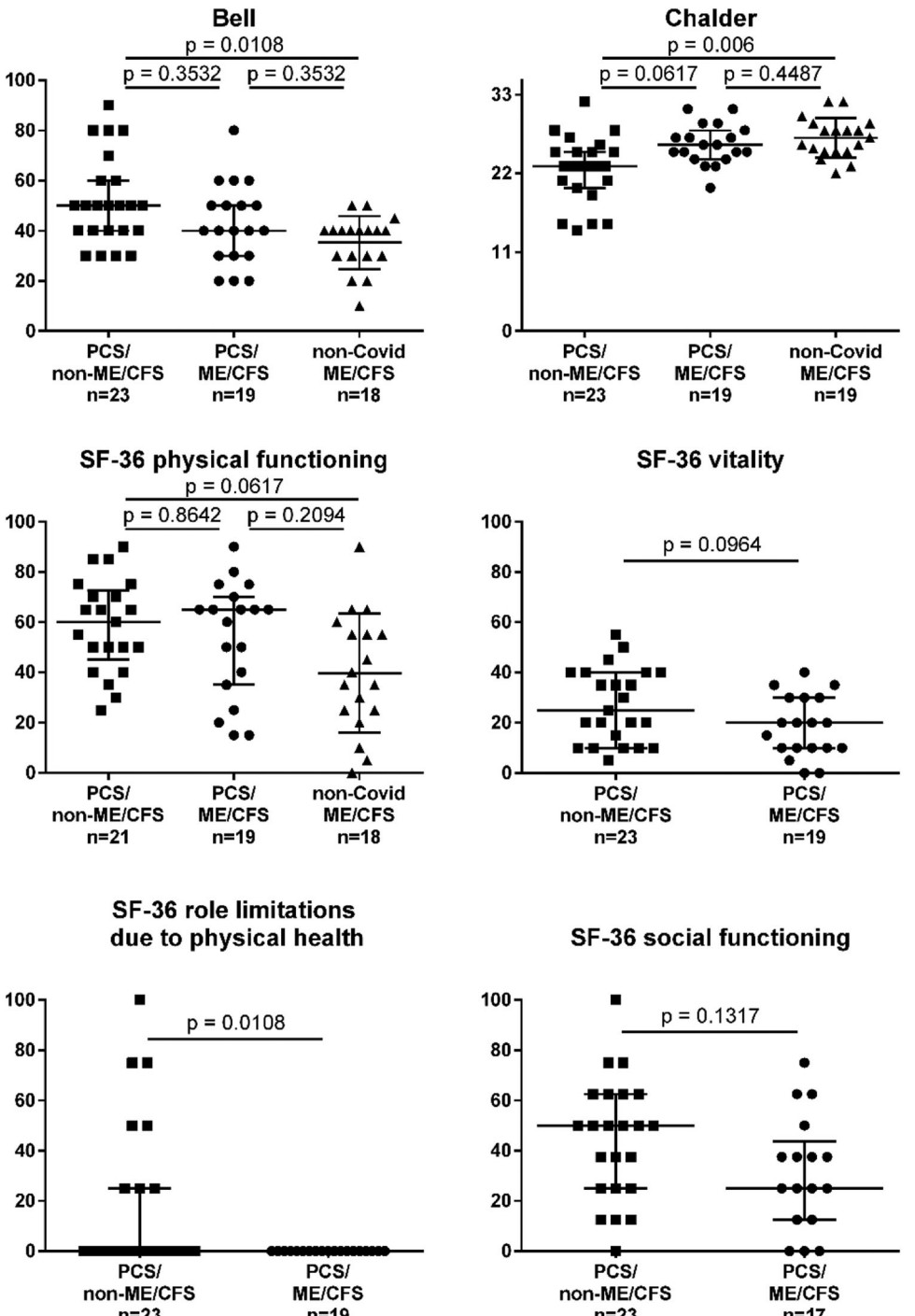

**Fig. 1 | Severity of fatigue and disability.** PCS/non-ME/CFS post-COVID-19 syndrome/non-myalgic encephalomyelitis/chronic fatigue syndrome; PCS/ME/CFS post-COVID-19 syndrome/myalgic encephalomyelitis/chronic fatigue syndrome; non-COVID ME/CFS non-COVID-19 myalgic encephalomyelitis/chronic fatigue syndrome; PEM post-exertional malaise; CFQ Chalder fatigue scale; SF-36 36-Item Short Form Survey. Bell disability scale (score 0–100), CFQ (score 0–33), and SF-36 physical function, vitality, role limitations, and social function (scores 0–100) were assessed in PCS (post-COVID-19 syndrome) and non-COVID-ME/CFS cohorts by questionnaires. Data were analyzed with nonparametric all-pairs Dunn-type multiple contrast tests (Bell disability score, CFQ, SF-36 physical functioning) and Brunner-Munzel tests. The *p* values were adjusted for multiplicity across endpoints with the Benjamini-Hochberg (BH) correction. Median are shown with IQR (interquartile range). Source data are provided as a Source Data file.

disability score of 40 and 30 out of 100, respectively (Fig. 1). According to the Bell disability scale, patients with a score of 30–40 are able to perform light work 2–4 h a day, thus requiring a reduced work schedule or are unable to work. Patients with a Bell disability score of 20 are confined to bed most of the day[22]. PCS/non-ME/CFS patients had a median Bell Score of 50 and a higher SF-36 sub-score for role limitations compared to PCS/ME/CFS patients (Fig. 1).

## Symptom severity

Fatigue as the leading symptom of PCS was assessed by the Chalder Fatigue Score (CFQ)[23]. PCS/non-ME/CFS patients reported less fatigue compared to non-COVID-19 ME/CFS (Fig. 1). As expected due to the diagnostic criteria frequency and severity of PEM as the cardinal symptom of ME/CFS was a strong discriminatory factor between PCS/non-ME/CFS and non-COVID ME/CFS patients, but differences were

not significant between PCS/non-ME/CFS and PCS/ME/CFS. While according to diagnostic criteria all ME/CFS patients had PEM of 14 h or more, 18 of 23 PCS/non-ME/CFS patients reported PEM of <14 h as shown in Fig. 2. PEM severity, frequency and length was similar in the PCS/ME/CFS and non-COVID-19 ME/CFS patients.

Table 2 displays the frequency and severity of key symptoms of the Canadian Consensus Criteria (CCC) quantified using a 1–10 scale. Post-exertional malaise (PEM), fatigue, stress intolerance, and hypersensitivity to temperature were less severe in PCS/non-ME/CFS compared to non-COVID ME/CFS and flu-like symptoms were less severe in both PCS cohorts compared to non-COVID ME/CFS following Benjamini-Hochberg (BH) correction. When only comparing the two PCS cohorts the higher symptom burden for stress intolerance and hypersensitivity to temperature, noise, and light in the PCS/ME/CFS cohort was significant, too (see Supplementary Fig. S1).

### Autonomic dysfunction
The majority of PCS patients suffered from autonomic dysfunction assessed by Composite Autonomic Symptom Score (COMPASS 31) with moderate symptoms (defined as a range between 20 and 40 out of 100) symptoms in 21 and severe symptoms (defined as a range of more than 40 from 100) in 11 patients[24]. Severity of symptoms was not significantly different between the cohorts. The COMPASS 31 total score and sub-scores of orthostatic, gastrointestinal, vasomotor, pupillomotor, secretory, and bladder symptoms are listed in Table 3.

### Hand grip strength (HGS)
Muscle fatigue and fatigability were assessed by ten repeat hand grips at maximum force (Fmax1 and Fmean1) and were repeated after 60 min (Fmax2 and Fmean2). Compared to reference values for age-matched healthy subjects[25], most patients were below the cut-off values for Fmax1/2 and Fmean1/2 discriminating healthy controls from ME/CFS[25] as shown in Fig. 3 for the female patients. Differences between cohorts were not found after BH correction.

### Sitting and standing heart rate and blood pressure
Heart rate as well as systolic and diastolic blood pressure sitting, standing, and after 2, 5, and 10 min standing was assessed in PCS/non-ME/CFS and PCS/ME/CFS patients and is shown in Fig. 4. Three female patients with PCS/non-ME/CFS and seven with PCS/ME/CFS had a sitting blood pressure of >140 mmHg systolic and/or >90 mmHg diastolic. Four patients (three female, one male) with PCS/ME/CFS and one patient with PCS/non-ME/CFS fulfilled diagnostic criteria for postural tachycardia syndrome (POTS)[26,27]. Orthostatic hypotension was diagnosed in six patients with PCS/ME/CFS (five females and one male) and one with PCS/non-ME/CFS (as shown in Fig. 4 for female patients).

### Laboratory parameters
Table 4 lists laboratory values in PCS/ME/CFS and PCS/non-ME/CFS patients, which were out of normal range in a subset of patients. Remarkably, mannose binding lectin (MBL) deficiency, which is associated with enhanced susceptibility to infections and was found previously more frequently in ME/CFS (15%) than in a historical control group (6%)[28], was also more frequent in both PCS patient cohorts with 17% and 23%[28]. While C-reactive protein (CRP) was slightly elevated in two PCS patients only, another marker of inflammation interleukin 8 (IL8), which we assessed in erythrocytes, was above the normal value in 37% and 48% of PCS/ME/CFS and PCS/non-ME/CFS patients, respectively, indicating inflammation during the last 3–4 months[29]. Elevated antinuclear antibodies (ANA) of 1:160–1:1280 were found in eight patients. Double-stranded DNA and extractable nuclear antigen antibodies (ENA) were negative in all patients and there was no evidence for a rheumatological disease. Four patients (two of each group) showed elevated anti-thyroid peroxidase (TPO) antibodies with

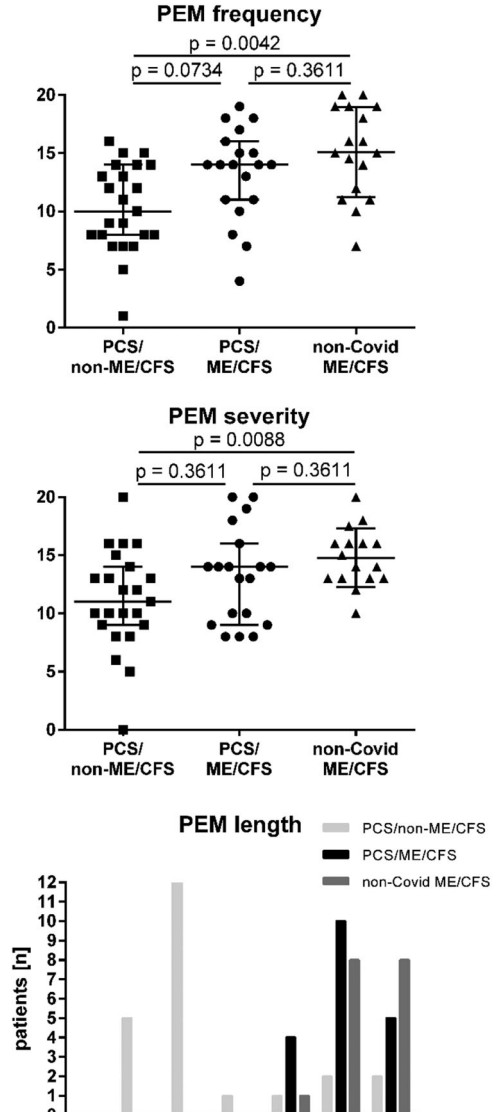

**Fig. 2 | Frequency, severity, and length of post-exertional malaise (PEM).** PCS/non-ME/CFS post-COVID-19 syndrome/non-myalgic encephalomyelitis/ chronic fatigue syndrome; PCS/ME/CFS post-COVID-19 syndrome/myalgic encephalomyelitis/chronic fatigue syndrome; non-COVID ME/CFS non-COVID-19 myalgic encephalomyelitis/chronic fatigue syndrome; PEM post-exertional malaise. Frequency and severity of PEM was assessed on a five items scale with 0–20 points ("none" to "all of the time"/"very severe") and the length in seven categories (from <1 h to 2–3 days), according to Cotler[10]. Median and IQR (interquartile range) in PCS/ME/CFS (n = 19), PCS/non-ME/CFS (n = 23), and non-COVID ME/CFS patients (n = 17 for frequency and length, and n = 16 for severity) is shown. Data were analyzed with nonparametric all-pairs Dunn-type multiple contrast tests. The p values were adjusted for multiplicity across endpoints with the Benjamini-Hochberg (BH) correction. Source data are provided as a Source Data file.

normal thyroid stimulating hormone (TSH) and free triiodothyronine/ thyroxine (fT3/fT4). Three of them had been diagnosed with Hashimoto thyroiditis before COVID-19. Deficiencies of vitamin D were found in 11% and 22% of PCS/ME/CFS and PCS/non-ME/CFS patients, respectively, and folic acid deficiencies in 19% of all patients. N-terminal prohormone of brain natriuretic peptide (NT-proBNP) was slightly elevated in three patients. Angiotensin converting enzyme 1 (ACE1) and ACE2 were assessed due to presumed dysregulation in PCS.

**Table 2 | Frequency and severity of symptoms**

| | PCS/non-ME/CFS (n = 23) | | | PCS/ME/CFS (n = 19) | | | non-COVID ME/CFS (n = 17) | | | non-COVID ME/CFS vs. PCS/non-ME/CFS | non-COVID ME/CFS vs. PCS/ME/CFS | PCS/non-ME/CFS vs. PCS/ME/CFS |
|---|---|---|---|---|---|---|---|---|---|---|---|---|
| | %PCS patients | Median | Range | % PCS/CFS patients | Median | Range | % ME/CFS patients | Median | Range | p | P | p |
| Fatigue | 100 | 7 | (2–10) | 100 | 8 | (5–10) | 100 | 8 | (5–10) | 0,05 | n.s. | n.s. |
| PEM | 100 | 6 | (1–9) | 100 | 8 | (5–10) | 100 | 9 | (7–10) | *** | n.s. | n.s. |
| Need for rest | 96 | 7[a] | (2–10) | 100 | 8 | (5–10) | 94 | 8,5[a] | (7–10) | n.s. | n.s. | n.s. |
| Impaired performance | 96 | 8[a] | (3–10) | 100 | 8 | (4–10) | 100 | 8 | (5–10) | n.s. | n.s. | n.s. |
| Stress intolerance | 96 | 6[a] | (2–10) | 100 | 8 | (3–10) | 94 | 9[a] | (8–10) | *** | n.s. | n.s. |
| Muscle pain | 83 | 4[a] | (1–9) | 84 | 5 | (1–10) | 88 | 7 | (1–10) | n.s. | n.s. | n.s. |
| Headache | 87 | 5 | (1–10) | 95 | 5[a] | (1–9) | 94 | 7[a] | (4–9) | n.s. | n.s. | n.s. |
| Joint pain | 78 | 3[a] | (1–9) | 89 | 3 | (1–10) | 76 | 4 | (0–8) | n.s. | n.s. | n.s. |
| Memory/word finding problems | 70 | 5[a] | (1–8) | 100 | 5 | (2–7) | 88 | 5[a] | (1–8) | n.s. | n.s. | n.s. |
| Concentration impairment | 91 | 5 | (1–9) | 100 | 6 | (3–9) | 100 | 7 | (4–9) | n.s. | n.s. | n.s. |
| Mental fatigue | 100 | 7 | (2–10) | 100 | 6 | (4–10) | 100 | 8 | (5–10) | n.s. | n.s. | n.s. |
| Visual disturbances | 48 | 1.5[a] | (1–6) | 63 | 3 | (1–6) | 71 | 3 | (1–9) | n.s. | n.s. | n.s. |
| Palpitations | 70 | 4[a] | (1–9) | 89 | 5 | (1–10) | 94 | 5 | (0–10) | n.s. | n.s. | n.s. |
| Dizziness when standing up | 83 | 4[a] | (1–8) | 84 | 5 | (1–10) | 94 | 5 | (1–10) | n.s. | n.s. | n.s. |
| Dizziness when walking | 61 | 2.5[a] | (1–9) | 68 | 3.5[a] | (1–7) | 88 | 4 | (1–10) | n.s. | n.s. | n.s. |
| Sleep disturbances | 83 | 6.5[a] | (1–10) | 89 | 7 | (1–10) | 94 | 8 | (1–10) | n.s. | n.s. | n.s. |
| Hypersensitivity to temperature | 48 | 1.5[a] | (1–8) | 79 | 5 | (1–8) | 88 | 7 | (1–10) | ** | n.s. | n.s. |
| .... to light | 52 | 2[a] | (1–7) | 84 | 5 | (1–10) | 88 | 5 | (1–10) | n.s. | n.s. | n.s. |
| .... to noise | 70 | 3[a] | (1–9) | 89 | 5 | (1–10) | 88 | 7 | (1–10) | n.s. | n.s. | n.s. |
| Breathing difficulty | 70 | 5 | (1–10) | 79 | 5 | (1–10) | 47 | 1.5[a] | (1–8) | n.s. | n.s. | n.s. |
| Irritable bowel | 48 | 2[b] | (1-10) | 79 | 5 | (1–9) | 82 | 6 | (1–10) | n.s. | n.s. | n.s. |
| Fever | 17 | 1 | (1–3) | 21 | 1[a] | (1–10) | 35 | 1[a] | (1–5) | n.s. | n.s. | n.s. |
| Painful lymph nodes | 30 | 1 | (1–7) | 32 | 1 | (1–9) | 71 | 3.5[a] | (1–7) | n.s. | n.s. | n.s. |
| Sore throat | 57 | 2 | (1–7) | 63 | 3 | (1–7) | 82 | 6 | (1–9) | n.s. | n.s. | n.s. |
| Flu-like symptoms | 70 | 3[a] | (1–10) | 79 | 5 | (1–8) | 100 | 8 | (4–9) | ** | * | n.s. |

Symptom severity was assessed on a scale of 1–10 (none to most severe) in PCS cohorts and non-COVID ME/CFS. Data were analyzed using nonparametric all-pairs Dunn-type multiple contrast tests.
*P* values were adjusted for multiplicity across symptoms with the Benjamini-Hochberg (BH) correction, statistically significant comparisons are indicated by asterisks as: *<0.05; **<0.01; ***<0.0001, not significant comparisons are indicated as n.s. Source data are provided as a Source Data file.
*PCS/non-ME/CFS* post-COVID-19 syndrome/non-myalgic encephalomyelitis/chronic fatigue syndrome, *PCS/ME/CFS* post-COVID-19 syndrome/myalgic encephalomyelitis/chronic fatigue syndrome, *non-COVID ME/CFS* non-COVID-19 myalgic encephalomyelitis/chronic fatigue syndrome.
[a]$n = n_{total}-1$.
[b]$n = n_{total}-2$.

Indeed, ACE1 levels were below the normal range in 31% of all patients. There were no differences between laboratory findings in PCS/ME/CFS and PCS/non-ME/CFS patients.

To investigate a potential pathophysiological role of laboratory parameters of relevance for oxygen supply, inflammation, and vasoregulation, we analyzed associations with levels of fatigue assessed by questionnaires and muscle fatigue determined by HGS within a correlation analysis using spearman's ρ (Fig. 5). Remarkably, we found several significant correlations following BH correction. HGS parameters showed a positive correlation with levels of hemoglobin in both PCS/non-ME/CFS and PCS/ME/CFS. Further we observed a negative correlation of Fmax1 with IL8 in erythrocytes and of Fmean2 with CRP and a positive correlation with ACE2 levels in the PCS/non-ME/CFS cohort. In the PCS/ME/CFS cohort there was a positive correlation of HGS parameters with bilirubin and ferritin and a negative correlation with NT-proBNP levels. In non-COVID ME/CFS patients we observed a similar correlation of bilirubin with HGS which was not significant after BH correction.

## Discussion

In this study, we provide evidence that a subset of PCS patients presenting with the hallmark of moderate to severe fatigue and exertion intolerance fulfill the CCC for ME/CFS[18]. In PCS patients who did not fulfill these criteria, this was mostly due to shorter duration and less severe PEM[10]. Postural tachycardia and hypotension were noted more frequently in PCS/ME/CFS, which was not unexpected as orthostatic symptoms are a hallmark of ME/CFS. A less severe symptom burden of the PCS/non-ME/CFS cohort was found in comparison to non-COVID ME/CFS patients with less fatigue, stress intolerance, hypersensitivity to temperature and flu-like symptoms. When only comparing the two PCS cohorts the higher symptom burden for stress intolerance and hypersensitivity to temperature, noise, and light in the PCS/ME/CFS cohort was significant, too (see Supplementary Fig. S1). Severity of symptoms was similar in the PCS/ME/CFS compared to the non-COVID-19 ME/CFS patient cohort with the only exception of less severe flu-like symptoms. A possible confounder might be the longer disease duration in the latter group.

**Table 3 | COMPASS 31 total score and subdomains**

| COMPASS 31 | PCS/non-ME/CFS (n = 23) | | PCS/ME/CFS (n = 19) | | non-COVID ME/CFS (n = 19) | |
|---|---|---|---|---|---|---|
| | **Median** | **Range** | **Median** | **Range** | **Median** | **Range** |
| Total | 26.8 | (2.5–54.0) | 39.4 | (7.1–62.2) | 41.0 | (7.8–66.4) |
| Orthostasis | 16.0 | (0–40) | 24.0 | (0–40) | 28.0 | (0–40) |
| Vasomotor | 0.0 | (0–4.2) | 0.0 | (0–4.2) | 0.0 | (0–4) |
| Secretomotor | 2.1 | (0–10.7) | 4.3 | (0–12.9) | 4.3 | (0–12.9) |
| Gastrointestinal | 6.3 | (0–16.1) | 6.3 | (0–15.2) | 8.9 | (3.6–14.2) |
| Bladder | 0.0 | (0–3.3) | 0.0 | (0–3.3) | 1.1 | (0–3.3) |
| Pupillomotor | 1.3 | (0–3.0) | 1.7 | (0–3.3) | 1.8 | (0–3.3) |

Autonomic symptoms were assessed by COMPASS 31 questionnaire (Composite Autonomic Symptom Score 31) in PCS cohorts and non-COVID-ME/CFS, considering the total score (0–100) and the scores of the six subdomains orthostasis (0–40), vasomotor (0–5), secretomotor (0–15), gastrointestinal (0–25), bladder (0–10) and pupillomotor (0–5). Data were analyzed using nonparametric all-pairs Dunn-type multiple contrast tests. The *p* values were adjusted for multiplicity across domains with the Benjamini-Hochberg (BH) correction, none of the comparisons remained significant. Source data are provided as a Source Data file.

*PCS/non-ME/CFS* post-COVID-19 syndrome/non-myalgic encephalomyelitis/chronic fatigue syndrome, *PCS/ME/CFS* post-COVID-19 syndrome/myalgic encephalomyelitis/chronic fatigue syndrome, *non-COVID ME/CFS* non-COVID-19 myalgic encephalomyelitis/chronic fatigue syndrome.

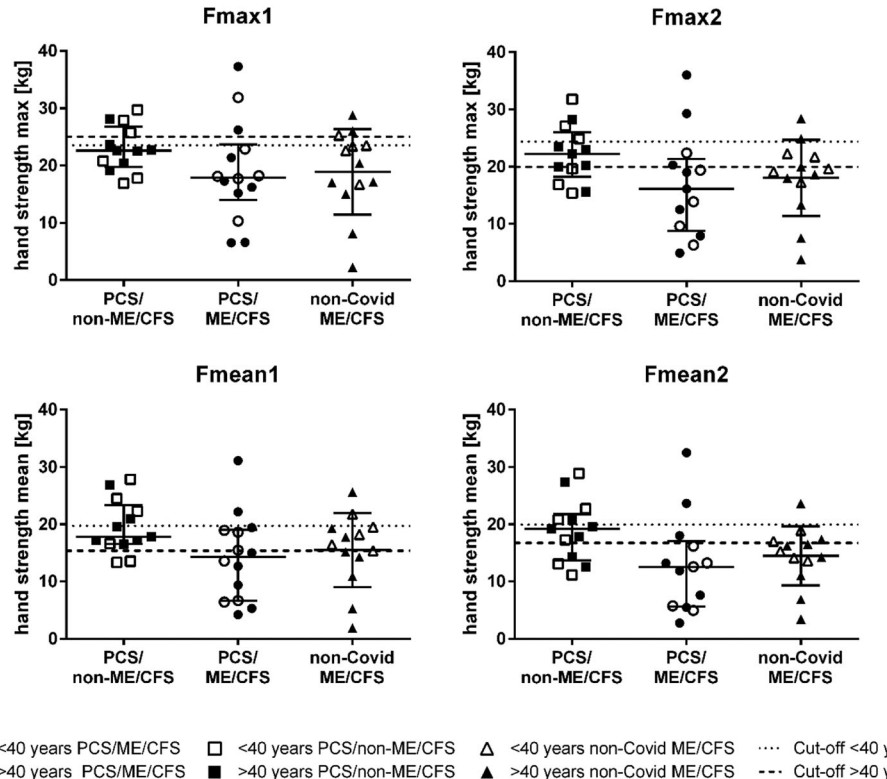

**Fig. 3 | Hand grip strength (HGS).** HGS hand grip strength; PCS/non-ME/CFS post-COVID-19 syndrome/non-myalgic encephalomyelitis/chronic fatigue syndrome; PCS/ME/CFS post-COVID-19 syndrome/myalgic encephalomyelitis/chronic fatigue syndrome; non-COVID ME/CFS non-COVID-19 myalgic encephalomyelitis/chronic fatigue syndrome. HGS was assessed in PCS cohorts (PCS/non-ME/CFS n = 13, PCS/ME/CFS n = 14 for Fmax1 and Fmean1, n = 13 for Fmax2 and Fmean2) and non-COVID-ME/CFS (n = 13). Fmax1 and Fmean1 of ten pulls in female patients and repeat assessment after 60 min (Fmax2 and Fmean2 of ten pulls) are shown (median and IQR (interquartile range)). Cut-off values of AUC reference values for age-matched healthy females are displayed as dashed lines: <40 years black dots and narrower dashed lines; >40 years white dots and wider dashed lines. Data were analyzed with nonparametric all-pairs Dunn-type multiple contrast tests. The *p* values were adjusted for multiplicity across endpoints with the Benjamini-Hochberg (BH) correction. Source data are provided as a Source Data file.

Several diagnostic criteria have been proposed for use in ME/CFS, of which CCC are recommended for diagnosis confirmation in secondary care and in research[30]. Severity and duration of PEM is a key diagnostic criterion of the CCC. In contrast to the original minimum length of 24 h of PEM required by the CCC, we set the duration criterion at 14 h as shown by others to yield the highest diagnostic sensitivity and specificity to discriminate patients with ME/CFS from patients with fatigue due to other chronic illnesses[10,14]. Strikingly, while all PCS patients suffered from moderate to severe fatigue and exertion intolerance, a subset did not fulfill the CCC criteria for ME/CFS mostly due to a shorter PEM lasting <14 h. We have not seen such a symptom constellation in other chronic postinfectious syndromes so frequently. The previously widely used Fukuda or CDC-1994 criteria do not require PEM for the diagnosis of ME/CFS thus most PCS/non-ME/CFS patients from our study would have been classified as ME/CFS[31,32]. Fukuda criteria are, however, no longer recommended to be used for ME/CFS diagnosis as they do not require PEM, the key symptom of ME/CFS[12,13]. The IOM (Institute of Medicine) criteria do not define the length of

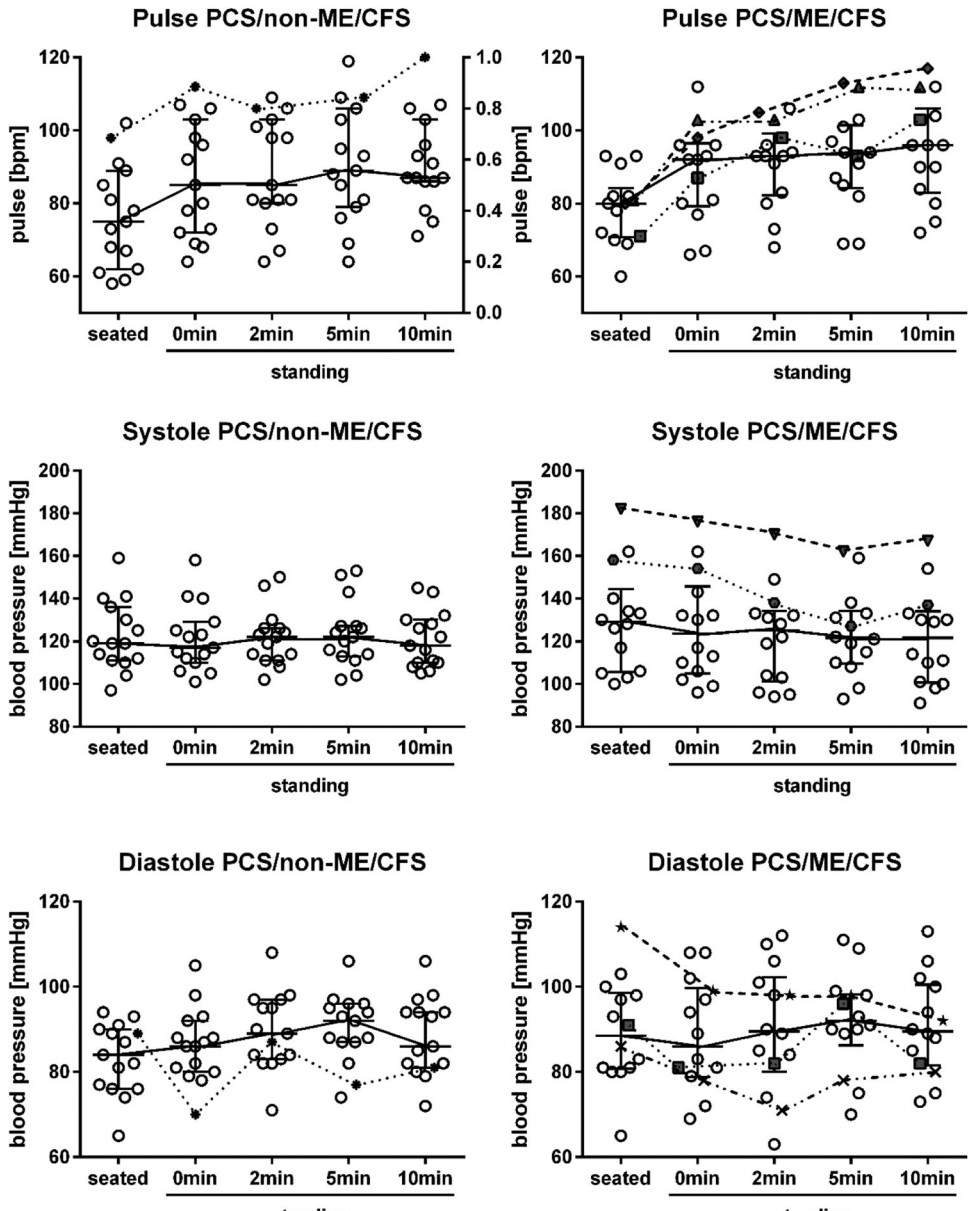

**Fig. 4 | Sitting and standing heart rate and blood pressure in female patients.**
PCS/non-ME/CFS post-COVID-19 syndrome/non-myalgic encephalomyelitis/
chronic fatigue syndrome; PCS/ME/CFS post-COVID-19 syndrome/myalgic ence-
phalomyelitis/chronic fatigue syndrome; BP blood pressure; POTS postural
orthostatic tachycardia syndrome. Heart rate (pulse) as well as systolic (systole) and
diastolic (diastole) BP sitting, standing, and after 2, 5, and 10 min standing in
females with PCS/non-ME/CFS (n = 15) or PCS/ME/CFS (n = 14). Other symbols than
blank dots represent patients with POTS (three with PCS/ME/CFS and one with PCS/
non-ME/CFS) and/or orthostatic hypotension (five patients with PCS/ME/CFS and
one with PCS/non-ME/CFS). Median are shown with IQR (interquartile range).
Source data are provided as a Source Data file.

PEM but require that it should occur at least half of the time with
moderate to severe intensity. Thus many patients not classified as ME/
CFS in our study would not have fulfilled the IOM criteria either[30].

ME/CFS is a debilitating disease leading to vast social, economic,
and individual impairments[30]. People with ME/CFS have been strug-
gling for decades to be recognized as such as many physicians
are unfamiliar with diagnosing and treating this disease. Despite the
less severe phenotype of some symptoms in the PCS/non-ME/CFS
subgroup compared to ME/CFS, most of these patients were severely
impaired in daily life, too. Based on the average Bell disability
scores, about two thirds of all patients were forced to reduce working
hours or are unable to work. This finding is in accordance with a recent
report of a patient survey on long COVID patients seven months after
infection[4].

Health sequelae of long COVID-19 can vary including post-
intensive care syndrome, pulmonary impairment, neurological defi-
cits, and posttraumatic stress disorder among others. In our patient
cohort of younger patients with mostly mild COVID-19, we have,
however, no evidence of potentially confounding organ impairment or
major depressive or anxiety diseases in accordance with other
reports[33,34]. A study from a pulmonary center reported that patients
with normal lung function three months after recovery from acute
mild COVID-19 exhibited more fatigue and more impairment of phy-
sical functioning and quality of life than patients who had moderate-to-
critical COVID-19[34]. Furthermore, in this study, only a minority of
patients showed evidence for depression or anxiety which is in line
with our data and provides evidence that despite a high illness burden
mental health is not relevantly impaired in most patients with PCS.

**Table 4 | Laboratory values in PCS cohorts**

| | | PCS/ME/CFS | | | | | | PCS/non-ME/CFS | | | | | | Reference range | |
|---|---|---|---|---|---|---|---|---|---|---|---|---|---|---|---|
| | Unit | n | Median | Min | Max | % low | % high | n | Median | Min | Max | % low | % high | m | f |
| CD4 Tcells abs | /nl | 19 | 0.75 | 0.31 | 2.07 | 5 | 11 | 23 | 0.72 | 0.31 | 1.52 | 17 | 4 | 0.5–1.2 | |
| CD8 Tcells abs | /nl | 19 | 0.41 | 0.21 | 0.97 | 16 | 11 | 23 | 0.37 | 0.21 | 0.61 | 35 | 0 | 0.3–0.8 | |
| Erythrocytes | /pl | 19 | 4.5 | 4 | 5 | 0 | 0 | 23 | 4.7 | 4 | 5.4 | 0 | 0 | 4.3–5.8 | 3.9–5.2 |
| Hemoglobin | g/dl | 19 | 13.7 | 12.2 | 15.4 | 0 | 0 | 23 | 13.7 | 12.2 | 17.6 | 0 | 4 | 13.5–17.0 | 12.0–15.6 |
| Thromboyctes | /nl | 19 | 263 | 143 | 376 | 5 | 5 | 23 | 233 | 168 | 452 | 0 | 4 | 150–370 | |
| Ferritin | µg/l | 19 | 91.9 | 16.7 | 337.2 | 0 | 16 | 21 | 69.9 | 11.2 | 235.1 | 5 | 5 | 30–400 | 13–150 |
| Creatinine | mg/dl | 18 | 0.785 | 0.61 | 1.04 | 0 | 22 | 21 | 0.77 | 0.61 | 0.99 | 5 | 5 | 0.7–1.2 | 0.5–0.9 |
| Creatine kinase | U/l | 19 | 74 | 25 | 152 | – | 0 | 21 | 61 | 41 | 273 | – | 5 | <190 | <167 |
| Lactate dehydrogenase | U/l | 19 | 205 | 147 | 317 | 0 | 16 | 22 | 195 | 124 | 280 | 5 | 5 | 135–250 | |
| Bilirubin | mg/dl | 18 | 0.505 | 0.22 | 1.6 | – | 6 | 22 | 0.49 | 0.2 | 0.97 | – | 0 | <1.2 | |
| GPT | U/l | 18 | 17.5 | 11 | 49 | – | 11 | 21 | 18 | 10 | 49 | – | 10 | <41 | <31 |
| GOT | U/l | 18 | 24.5 | 17 | 33 | – | 0 | 21 | 22 | 15 | 37 | – | 0 | <49 | <35 |
| NT-proBNP | ng/l | 18 | 47 | 7 | 125 | – | 11 | 21 | 39 | 6 | 181 | – | 5 | a | |
| ACE 1 | U/l | 17 | 26.8 | 10.3 | 52.5 | 29 | 0 | 22 | 24.85 | 12.2 | 37.6 | 32 | 0 | 20–70 | |
| ACE 2 | ng/ml | 18 | 4.05 | 2.5 | 541.7 | 6 | 17 | 20 | 4.6 | 2.7 | 387.6 | 0 | 25 | n.a. | |
| fT3 | ng/l | 18 | 3.325 | 2.54 | 3.98 | 0 | 0 | 23 | 3.27 | 2.57 | 4.08 | 0 | 0 | 2–4.4 | |
| fT4 | ng/l | 18 | 13.8 | 11 | 18.1 | 0 | 11 | 23 | 13.1 | 9.77 | 15.9 | 0 | 0 | 9.3–17 | |
| TSH basal | mU/l | 18 | 1.385 | 0.37 | 2.36 | 0 | 0 | 21 | 1.33 | 0.7 | 3.31 | 0 | 0 | 0.27–4.2 | |
| Anti-TPO Ab | kU/l | 15 | 9 | 9 | 328 | – | 13 | 19 | 9 | 9 | 81 | – | 11 | <34 | |
| IgG | g/l | 19 | 10.94 | 7.13 | 13.22 | 0 | 0 | 23 | 10.63 | 6.73 | 16.18 | 4 | 4 | 7–16 | |
| IgA | g/l | 19 | 1.75 | 1.01 | 4.09 | 0 | 11 | 23 | 1.92 | 0.32 | 4.62 | 4 | 4 | 0.7–4 | |
| IgM | g/l | 19 | 1.15 | 0.63 | 5.05 | 0 | 11 | 23 | 1.09 | 0.45 | 13.48 | 0 | 9 | 0.4–2.3 | |
| IgE | g/l | 19 | 46 | 2.4 | 1072 | 0 | 21 | 23 | 35.1 | 4 | 751 | 0 | 22 | 0–100 | |
| IgG1 | g/l | 19 | 5.797 | 3.637 | 7.226 | 0 | 0 | 22 | 6.034 | 3.13 | 8.209 | 0 | 5 | 2.8–8 | |
| IgG2 | g/l | 19 | 3.904 | 2.607 | 6.445 | 0 | 5 | 22 | 4.881 | 2.752 | 5.934 | 0 | 5 | 1.12–5.7 | |
| IgG3 | g/l | 19 | 0.473 | 0.15 | 1.342 | 5 | 5 | 22 | 0.505 | 0.19 | 1.062 | 5 | 0 | 0.24–1.25 | |
| IgG4 | g/l | 19 | 0.262 | 0.132 | 2.115 | 0 | 5 | 22 | 0.544 | 0.096 | 1.781 | 0 | 9 | 0.052–1.25 | |
| Complement C3 | mg/l | 18 | 1080 | 850 | 1640 | 11 | 0 | 22 | 1155 | 820 | 1720 | 14 | 0 | 900–1800 | |
| Complement C4 | mg/l | 18 | 185 | 100 | 330 | 0 | 0 | 22 | 210 | 110 | 320 | 0 | 0 | 100–400 | |
| MBL | ng/ml | 18 | 1452 | <50 | 4000 | 17 | – | 22 | 3734 | <50 | 4000 | 23 | – | >50 | |
| IL8 in erythrocytes | pg/ml | 19 | 134.8 | 97 | 224 | – | 37 | 23 | 149.2 | 65.2 | 442 | – | 48 | <150 | |
| Soluble IL2 receptor | IU/ml | 19 | 312 | 176 | 746 | – | 5 | 23 | 326 | 173 | 765 | – | 4 | <710 | |
| C-reactive protein | mg/l | 19 | 0.8 | 0.6 | 7.1 | – | 5 | 23 | 1 | 0.6 | 7.57 | – | 4 | <5 | |
| Vitamin D3 | nmol/l | 19 | 80.4 | 46.2 | 109.6 | 11 | 0 | 23 | 62.4 | 16.7 | 217.6 | 22 | 4 | 50–150 | |
| Folic acid | µg/l | 16 | 7.3 | 2.5 | 20 | 19 | 13 | 21 | 9.7 | 2.7 | 20 | 19 | 5 | 4.6–18.7 | |

Abs absolute, ACE angiotensin converting enzyme 1, fT3/fT4 free triiodothyronine/thyroxine, Ig immunoglobulin, IL8 interleukin 8, MBL mannose binding lectin receptor, PCS/non-ME/CFS post-COVID-19 syndrome/non-myalgic encephalomyelitis/chronic fatigue syndrome, PCS/ME/CFS post-COVID-19 syndrome/myalgic encephalomyelitis/chronic fatigue syndrome, TSH thyroid stimulating hormone, TPO thyroid peroxidase.
a NT-proBNP reference values according to age: ≤44 years = ≤97, ≤54 years = ≤121, ≤64 years = ≤198. Source data are provided as a Source Data file.

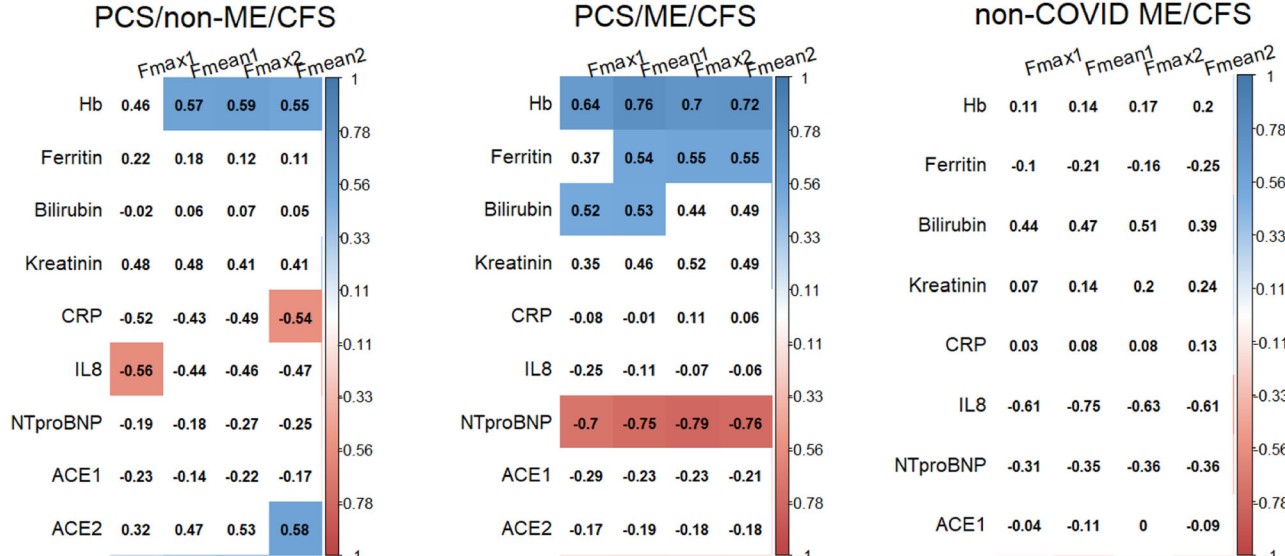

**Fig. 5 | Correlation of hand grip strength (HGS) with laboratory parameter.** PCS/non-ME/CFS post-COVID-19 syndrome/non-myalgic encephalomyelitis/chronic fatigue syndrome; PCS/ME/CFS post-COVID-19 syndrome/myalgic encephalomyelitis/chronic fatigue syndrome; non-COVID ME/CFS non-COVID-19 myalgic encephalomyelitis/chronic fatigue syndrome; Hb hemoglobin; CRP C-reactive protein; IL8 interleukin 8; NT-proBNP N-terminal prohormone brain natriuretic peptide; ACE angiotensin converting enzyme; HGS hand grip strength. Correlation of HGS parameter Fmax1, Fmean1, Fmax2, and Fmean2 with laboratory parameters of relevance for oxygen supply, inflammation, and vasoregulation. We analyzed the data within a correlation analysis using spearman's ρ and Benjamini-Hochberg (BH) correction for multiplicity. Correlations that were significant are indicated by blue (positive) or red (negative). Source data are provided as a Source Data file.

There is still no specific treatment for ME/CFS and knowledge of pathomechanisms is scarce and fragmented due to little interest and research support[35]. However, there is evidence of immune, autonomic, and metabolic dysregulation in postinfectious ME/CFS[15]. In line with these data, most patients in our study presented with symptoms of autonomic dysfunction. COVID-19 triggers a strong inflammatory response and there is evidence for autoimmunity triggered by COVID-19.[36] We have no indication for ongoing overt inflammation as only two of the patients presented with mildly elevated CRP. Almost half of the patients had, however, elevated IL-8 levels in erythrocytes. It has been shown that a high percentage of IL-8, mainly released by monocytes and endothelial cells, is stored in erythrocytes, which bind IL-8 via a duffy antigen receptor for chemokines[29,37,38]. Elevated ANA in eight patients (seven female/one male) and the preponderance of females may indicate an autoimmune mechanism similar to ME/CFS triggered by other infections[15–17]. MBL deficiency has been implicated in increased susceptibility to and severity of infections and was found more frequently in both cohorts in accordance with findings from a previous study in ME/CFS[28].

HGS is a reliable parameter to assess muscle fatigue and correlates with disease severity and PEM in ME/CFS[25]. Remarkably we found diminished HGS in the majority of patients and an association of several biomarker with muscle fatigue. Consistently with our clinical observations these associations may point to common and distinct pathomechanisms in PCS/non-ME/CFS and PCS/ME/CFS patients. First, positive correlations of HGS parameters with hemoglobin levels observed in both PCS cohorts, with ACE2 in the PCS/non-ME/CFS cohort, and with bilirubin and ferritin in the PCS/ME/CFS cohort as well as negative correlations with NT-proBNP levels in the PCS/ME/CFS cohort point to endothelial dysfunction and hypoperfusion as cause of muscle fatigue. All of these parameters may exert a protective function on endothelial function or muscle oxygen supply (hemoglobin oxygen supply), ACE2 (vasodilation), and bilirubin (vasodilation and antioxidant)[39]. NT-proBNP is a marker for heart failure and is released by distension of heart muscle cells. However, we had no evidence for impaired cardiac function in our patients. In addition, BNP is produced in ischemic skeletal muscle satellite cells as a potential paracrine

regulator of vasodilatation and vascular regeneration[40]. Thus, negative association of NT-proBNP levels with HGS is consistent with our hypothesis of hypoperfusion as cause of muscle fatigue. In the PCS/non-ME/CFS cohort, a negative correlation of Fmax1 with IL8 in erythrocytes and CRP levels with Fmean2 was found which may point to low level inflammation as mechanism of muscle fatigue. Endothelial dysfunction is considered as an important pathomechanism in ME/CFS[41–43]. Endothelial dysfunction and endothelitis are described in the pathogenesis of acute COVID-19. We also found endothelial dysfunction in a subset of both PCS cohorts[44]. A recent study reported elevated levels of circulating endothelial cells in COVID-19 convalescents as a biomarker for endothelial dysfunction associated with levels of several cytokines[45]. In the non-COVID ME/CFS cohort analyzed in this study we observed a similar positive correlation of HGS parameters with bilirubin (not significant after BH correction). A possible explanation for the weaker correlations might be longer disease duration of the patients in the ME/CFS cohort (13 months) and several different infectious triggers.

Limitations of our study are the lack of control groups of post-COVID-19 patients without fatigue, SARS-CoV-2-negative healthy persons and patients with fatigue unrelated to COVID-19. The non-COVID ME/CFS group is more heterogeneous in terms of longer disease durations and several different infectious triggers. Further, PCS patients studied represent a subgroup of patients selected for moderate to severe fatigue and exertion intolerance, so we do not know how representative they are. We will be able to answer this question from an ongoing cohort study at Charité of 300 randomly selected patients with a positive SARS-CoV-2 PCR test in March 2020. A recent study from Italy provides a better estimate of the frequency of ME/CFS in PCS[46]. The Italian group recruited all PCS patients seen at their clinic <65 years, >6 months follow-up after COVID-19 and without comorbidity. From these 37 patients 27% ($n = 10$) fulfilled the International Consensus Criteria for ME/CFS[47]. Furthermore, our study design and evaluation are purely exploratory with hypothesis generating character. By nature of the study, the results might be biased due to uncontrolled confounders[48]. Furthermore, note that the BH correction assumes

independent test results and *p* values and its application therefore is debatable. The Benjamini-Yekutieli correction turned out to be too conservative and we therefore sticked with BH. Since the study is exploratory, the adjustment helps in detecting initial findings.

Taken together, our study provides evidence that patients following mild COVID-19 develop a chronic syndrome fulfilling diagnostic criteria of ME/CFS in a subset. By defining and characterizing subgroups of PCS patients we could identify associations of HGS with biomarkers which may indicate hypoperfusion and inflammation as potential pathomechanisms. We must anticipate that this pandemic has the potential to dramatically increase the number of ME/CFS patients. At the same time, it offers the unique chance to identify ME/CFS patients in a very early stage of disease and apply interventions such as pacing and coping early with a better therapeutic prognosis. Further, it is an unprecedented opportunity to understand the underlying pathomechanism and characterize targets for specific treatment approaches.

## Methods

All patients signed informed consent before study inclusion. This study is part of the Pa-COVID-19 study of the Charité[19] and approved by the Ethics Committee of the Charité - Universitätsmedizin Berlin in accordance with the 1964 Declaration of Helsinki and its later amendments (EA2/066/20).

### Cohort and study protocol

The primary objective of this monocentric prospective observational cohort study was to characterize patients contacting the Charité Fatigue Center with persistent fatigue and exertion intolerance after COVID-19 and determine if they fulfill diagnostic criteria for ME/CFS. All patients enrolled in this study presented at our outpatient clinics between August 2020 and November 2020. We informed patients on our website that our outpatient clinic offers a study for patients suffering from moderate to severe fatigue and exertion intolerance three to six months post-COVID-19. Patients were selected for an appointment in our clinic based on a screening questionnaire, which specified our inclusion criteria. From a total of 81 patients, who contacted the Charité Fatigue Center during this time period, 24 were already excluded based on the screening questionnaire and 57 were assessed for eligibility in our outpatient clinic (see Fig. S2 for Consort flow diagram). From these, 42 fulfilled the inclusion criteria: (1) confirmed diagnosis of mild to moderate COVID-19, (2) persistent moderate to severe fatigue and exertion intolerance postinfection, and (3) absence of relevant cardiac, respiratory, neurological, or psychiatric comorbidity. Patients were thus excluded from this study in case of relevant comorbidities or preexisting fatigue, or evidence of organ dysfunction. For comorbidities, we refer to a list of diseases, in which fatigue may be a prominent feature and which may preclude a diagnosis of ME/CFS according to the European Network on ME/CFS (EUROMENE) guidelines[30]. All patients had to provide proof of previous COVID-19 diagnosis by positive SARS-CoV-2-polymerase chain reaction (PCR) or serology (anti-SARS-CoV-2-IgG). Further three patients without any PCR and with negative serology were included due to typical initial symptoms of loss of smell and taste due to the high diagnostic specificity of these symptoms in accordance with the inclusion criteria of our study protocol[49]. The participation in the study has not been financially compensated.

In all patients, neurological, pulmonary, and cardiac diseases were excluded by the respective specialist either before referral to our fatigue outpatient clinic or in our clinic. This included electrocardiogram, echocardiogram, chest X-ray and pulmonary function test in all patients. All patients included in the study were seen at our clinic by a rheumatologist to exclude a rheumatologic disease. In patients who reported moderate to severe difficulties with breathing, chest computer tomography (CT) and pulmonary function tests including

diffusion capacity were performed additionally. Patients with impaired diffusion capacity were not included in the study. Patients who reported severe cognitive impairment or severe headache received a detailed comprehensive standardized neurological assessment by a neurologist. Patients with sitting or postural tachycardia, a history of chest pain or palpitations or elevated NT-proBNP received a further examination by a cardiologist including 24 h electrocardiogram and echocardiography. Liver and renal dysfunction was excluded based on normal values including glomerular filtration rate.

All patients had COVID-19 between March and June 2020. During this time, there were no variants of SARS-CoV-2 reported in our region. We report here on the results of cross-sectional analyses at month six after onset of COVID-19 in a total of 42 patients. From all post-infectious non-COVID ME/CFS patients evaluated during the same period at our clinic (*n* = 123) a sex- and age-matched control cohort who had the shortest duration of illness (13 months, range 7–19 months, *n* = 19) was selected.

### Diagnostic criteria for ME/CFS and symptom assessment

Severity of mental and physical fatigue was assessed using the Chalder Fatigue Scale (CFQ)[23]. The first seven questions assess mental fatigue (CFQA), the last four physical fatigue (CFQB). Disability and daily physical function were assessed by the Bell disability scale and Short Form Health Survey-36 (SF-36 Version 1)[22,50]. The Bell disability scale is scored from 0 (very severe, bedridden constantly) to 100 (healthy)[22]. Frequency, severity, and duration of PEM symptoms were assessed according to Cotler et al.[10]. Symptoms of autonomic dysfunction were assessed by the Composite Autonomic Symptom Score (COMPASS 31)[24]. Depression and sleepiness were assessed by the Patient Health Questionnaire 9 (PHQ9) and the Epworth Sleepiness Scale ESS[20,21]. According to PHQ9, patients were classified as having minimal (1–4), mild (5–9), moderate (10–14), moderately severe (15–19), or severe depressive symptoms (20–27)[20]. According to ESS, patients were classified as no evidence of sleep apnea (0–9), possible mild to moderate sleep apnea (11–15), or possible severe sleep apnea (>16)[21].

Diagnosis of ME/CFS was based on Canadian Consensus Criteria (CCC) and exclusion of other diseases, which may be considered as potential confounding comorbidities serving as an alternative explanation for chronic fatigue[18]. In contrast to the original classification and in accordance with the studies of Lenard Jason and his team, a minimum of 14 h of PEM instead of 24 h was required for diagnosis of ME/CFS[10]. In addition, key symptoms of CCC were quantified using a 1–10 scale to assess severity of symptoms. All data were recorded using the Research Electronic Data Capture (REDCap) database. Supplementary material shows complete questionnaires, criteria, and assessments with interpretation.

### Functional studies, imaging and laboratory values

Hand grip strength (HGS) was assessed using an electric dynamometer assessing maximal and mean force of maximal pulls (Fmax1 and Fmean1) repeated ten times and a second assessment 60 min later (Fmax2 and Fmean2)[25]. Blood pressure and heart rate were assessed in sitting position as well as in standing position immediately after standing up and after 2, 5, and 10 min. Postural orthostatic tachycardia syndrome (POTS) is defined as pulse increase of more than 30 bpm compared to sitting or over 120 bpm both within 10 min after standing up and signs of orthostatic intolerance[26,27]. Orthostatic hypotension is defined as a decrease of more than 20 mmHg of systolic or 10 mmHg of diastolic blood pressure compared to sitting[27]. Laboratory parameters including full (CBC) and differential (DBC) blood count, lymphocyte subsets, interleukin 8 (IL8) in erythrocytes, mannose-binding lectin (MBL), C-reactive protein (CRP), immunoglobulin subsets, antinuclear antibodies (ANA), extractable nuclear antigen (ENA), complement C3/4, anti-thyreoperoxidase (TPO) antibodies, thyroid-stimulating hormon (TSH), free triiodothyronine/thyroxine (fT3/4),

ferritin, creatinine, liver enzymes, angiotensin-converting enzyme 1/2 (ACE1/2), N-terminal prohormone of brain natriuretic peptide (NT-proBNP) were determined at the Charité diagnostics laboratory (Labor Berlin GmbH, Berlin, Germany). ACE2 was assessed by an enzyme-linked immunoassay (ELISA; R&D Systems).

### Patient and public involvement

A German Facebook group maintained by patients suffering from long COVID contacted us first in June 2020 sharing their stories and symptom observations (https://longcoviddeutschland.org/). Our study design was developed based on frequency, type, and severity of symptoms reported and discussed with the patient group. The possibility for local patients to participate in our study was communicated on their website.

### Statistical analysis

For descriptive purpose, we illustrate the outcomes of all variables using median and range (if not indicated otherwise). Inferentially, we analyzed the data using purely nonparametric all-pairs Dunn-type multiple contrast tests (accounting for variance heteroscedasticty)[51]. Two samples are compared with the Brunner-Munzel test. All methods used are ranking methods testing hypotheses formulated in terms of so-called relative effects (Wilcoxon-Mann-Whitney parameters). Effect estimators and their standard errors, test statistics, and 95% simultaneous confidence intervals are provided in the supplementary material. Furthermore, since the number of endpoints and comparisons is pretty large (and to provide conservative estimates), we additionally adjust $p$ values for multiplicity across endpoints with the Benjamini-Hochberg (BH) correction. We performed all statistical computations using the statistical software package R using the R-packages nparcomp[52] and the p.adjust function. Furthermore, we estimated Spearman's $\rho$ and illustrate the results from correlation analysis in correlation plots using the R-package corrplot. The amount of missing values is very low and we therefore used all-available cases for data analyses in the correlation analyses. All results are interpreted in an exploratory manner at 5% level of significance (two-sided).

### Reporting summary

Further information on research design is available in the Nature Research Reporting Summary linked to this article.

## Data availability

Source data are provided with this paper. Due to sensitive nature of the data pseudonymized patient data may be available upon written reasonable request to the corresponding author (C.K.). The protocol synopsis is available as supplementary files within the original submission. Source data are provided with this paper.

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

## Acknowledgements

We thank Silvia Thiel for patient care and data management. We thank Sebastian Lorenz for statistic calculation of multiple comparisons. We thank Franziska Sotzny for support with figures and revision of content. We thank all patients who gave us their consent to publish their data in this study. We thank all members of the Pa-COVID-19 collaborative study group. All relevant funding awarded are described. This work is supported by a grant from the Weidenhammer-Zoebele Foundation. The work of FK was supported by the VolkswagenStiftung. The lead author (the manuscript's guarantor) affirms that the paper is honest, accurate, and transparent; that no important aspects of the study have been omitted; and that any discrepancies from the study as planned have been explained. Dissemination to participants and related patient and public communities is encouraged by open access publication and citing the study on our site https://cfs.charite.de/. We are engaging with print and internet press, television, radio, news, and documentary program makers. All items of the STROBE checklist are covered in the paper.

## Author contributions

C.S.c., C.K., and F.P. developed the concept of the study. H.D.V. gave important input into study concept and objectives. H.F. and C.K. were responsible for data curation and analyses of data. F.K. performed statistical analyses. C.K., C.S.c., K.W., L.H., F.S., J.B.S., L.M.A., F.P., B.H., C.S.k., J.S., and T.B. were involved in clinical investigation. M.H. was involved in data transfer, patient care, and collection of further information for revision. R.G. were involved in data transfer and patient care. T.Z. was involved in ethical affairs and data management. H.F., C.K., and C.S.c. validated the data. H.F. was involved in data visualization. C.S.c wrote the original draft of the paper. C.K., H.F., and U.B. reviewed and edited the paper. C.K., H.F., J.B.S., and C.S.c. contributed equally. All authors revised and approved the paper. The corresponding author attests that all listed authors meet authorship criteria and that no others meeting the criteria have been omitted. C.S.c. is the guarantor.

## Funding

## Competing interests

The authors declare no competing interests.

### Ethical approval

Ethical approval was given by the Ethics Committee of Charité - Universitätsmedizin Berlin in accordance with the 1964 Declaration of Helsinki and its later amendments (EA2/066/20).

### Informed consent

All participants provided written informed consent.

## Additional information

Claudia Kedor [1,13] ✉, Helma Freitag[1,13], Lil Meyer-Arndt[2,3,4], Kirsten Wittke[1], Leif G. Hanitsch[1], Thomas Zoller [5], Fridolin Steinbeis[5], Milan Haffke [1], Gordon Rudolf[1], Bettina Heidecker[6], Thomas Bobbert[7], Joachim Spranger [7], Hans-Dieter Volk [1,8], Carsten Skurk [6], Frank Konietschke[9], Friedemann Paul[2,3,4], Uta Behrends[10,11,12], Judith Bellmann-Strobl [1,2,3,4,13] & Carmen Scheibenbogen[1,13]

[1]Charité - Universitätsmedizin Berlin, corporate member of Freie Universität Berlin and Humboldt Universität zu Berlin, Institute of Medical Immunology, Berlin, Germany. [2]Experimental and Clinical Research Center, a cooperation between the Max Delbrück Center for Molecular Medicine in the Helmholtz Association and Charité Universitätsmedizin Berlin, Berlin, Germany. [3]Experimental and Clinical Research Center, Charité – Universitätsmedizin Berlin, corporate member of Freie Universität Berlin and Humboldt-Universität zu Berlin, Berlin, Germany. [4]Max Delbrück Center for Molecular Medicine in the Helmholtz Association (MDC), Berlin, Germany. [5]Department of Infectious Diseases and Respiratory Medicine, Charité – Universitätsmedizin Berlin, Corporate Member of Freie Universität Berlin and Humboldt-Universität zu Berlin, Berlin, Germany. [6]Department of Cardiology, Charité - Universitätsmedizin Berlin, corporate member of Freie Universität Berlin and Humboldt Universität zu Berlin, Berlin, Germany. [7]Department of Endcrinology and Metabolism, Charité - Universitätsmedizin Berlin, corporate member of Freie Universität Berlin and Humboldt Universität zu Berlin, Berlin, Germany. [8]Center for Regenerative Therapies (BCRT), Berlin Institute of Health, Berlin, Germany. [9]Institute of Biometry and Clinical Epidemiology, Charité – Universitätsmedizin Berlin, corporate member of Freie Universität Berlin and Humboldt-Universität zu Berlin, Berlin, Germany. [10]Childrens' Hospital, School of Medicine, Technical University of Munich, Munich, Germany. [11]German Center for Infection Research (DZIF), Berlin, Germany. [12]AGV Research Unit Gene Vectors, Helmholtz Center Munich (HMGU), Munich, Germany. [13]These authors contributed equally: Claudia Kedor, Helma Freitag, Judith Bellmann-Strobl, Carmen Scheibenbogen. ✉e-mail: claudia.kedor@charite.de

