## [Peer Review File · Nature Communications]

Title: Post COVID-19 Chronic Fatigue Syndrome following the first pandemic wave in Germany and biomarkers associated with symptom severity: results from a prospective observational studyEditorial Note: This manuscript has been previously reviewed at another journal that is not operating a transparent peer review scheme. This document only contains reviewer comments and rebuttal letters for versions considered at *Nature Communications*.

REVIEWER COMMENTS

Reviewer #1 (Remarks to the Author):

[Note: CAPS are used to emphasize some words because the software does not allow the use of italics or underlining.]

Post-acute COVID-19 syndrome is an emerging and serious health problem, and manuscripts on this topic therefore are welcome.

However, this revised manuscript, transferred from Nature Medicine to Nature Communications, continues to have serious deficiencies. The authors have provided some of the critically important Methods detail that was absent in the first draft of the manuscript. However, as outlined below, four major problems remain.

FIRST PROBLEM. Two reviewers of the previous manuscript cited the importance of having a comparison group of people without COVID-19. The authors state, in their Rebuttal and in the Methods section of this revised manuscript (p. 19), that “A sex and age matched postinfectious non-COVID-19 ME/CFS group with a mean disease onset of 13 months (range 7 - 19 months) was seen and diagnosed during the same period in our outpatient clinic by the same physicians and was selected for comparison.” That is a very appropriate comparison group. However, it would also be valuable to include a HEALTHY comparison group (sex and age matched) that was seronegative for SARS-CoV-2.

SECOND PROBLEM. While I was delighted to read about the addition of the non-COVID-19 ME/CFS comparison group, I then was very disappointed. The purpose of suggesting a comparison group, obviously, was to compare the putative biomarker results in CCS and CCS/CFS to the results in the comparison group. But where are those results? It is absolutely essential that they be included in Tables 1-3, along with the values for CCS and CCS/CFS. Otherwise, we have little basis for believing that the putative biomarkers really do discriminate the patient groups from other patients who have not suffered from COVID-19.

THIRD PROBLEM. Tables 2 and 3 show many differences in the results between the CCS and CCS/CFS groups to be statistically significant. However, those tables contain multiple comparisons. Furthermore, when those tables include results from the non-COVID-19 ME/CFS comparison group (see Second Problem), they will include even more comparisons.

In such a situation, the p-value chosen to be “statistically significant” must be adjusted for multiple comparisons. Yet the authors state (p. 21) that “A p-value of <0.05 was considered as statistically significant. Due to multiple testing p-values are considered descriptive without adjustments for multiple comparisons.”

I read this statement to mean that the authors did not use any of the several widely-used techniques to adjust for multiple comparisons. In my opinion, the failure to adjust for multiple comparisons would be indefensible. Without this, the authors cannot claim that ANY of the biomarkers that they say reveal “significant” differences really do reveal such differences. Since such differences are the rationale for submitting this report for publication, this failure is a significant problem.

FOURTH PROBLEM. The authors examined a number of potential biomarkers to see if they discriminated between the CCS and CCS/CFS groups. Many of these biomarkers are commonly measured and useful in diagnosing disease (e.g., CRP, TSH, vitamin D level). However, others are not: e.g., mannose binding lectin, IL-8 concentration within erythrocytes, serum ACE-1 levels. The authors need to explain WHY these unusual markers were chosen, and what they think abnormalities in these markers say about the underlying biology of CCS.

There are several smaller issues.

Were the laboratory abnormalities reported on page 9 (for example, thyroid function tests indicating Hashimoto’s thyroiditis) NEW, or had they also been present prior to acute COVID-19? The reader needs to know this.

The authors should consider changing the name “Chronic COVID-19 Syndrome (CCS)” to “post-acute COVID-19 syndrome (PACS)”, a term that is being used increasingly. This term is preferred because we do not yet know how long the debilitating symptoms following COVID-19 will last: will they last for many years, as with other “chronic” illnesses? If not, the term “chronic” may be misleading. Also, at the first use of the term “post-acute COVID-19 syndrome”, it might be appropriate to add “(also known as ‘long COVID’)”.

Reviewer #2 (Remarks to the Author):

In this revised manuscript, the authors address many of the reviewer comments from the prior submission. However, the integration of a control cohort (COVID-19 negative ME/CFS patients) is somewhat incomplete, particularly in terms of figures and tables, and laboratory data is lacking. In addition, there is no addition of a fatigue negative COVID-19 positive cohort, which would allow for a more relevant investigation of disease mechanisms. Presence of similar abnormal laboratory values in a proper control group is a minimum step necessary to suggest a mechanistic connection rather than correlation. The authors suggest that fatigue in CCS/CSF has a different mechanism than fatigue in CCS,

but it is unclear in the conclusions whether CCS/CSF should be lumped with COVID-19 negative ME/CFS, or if three different pathophysiological entities are present.

Reviewer #3 (Remarks to the Author):

This is a prospective observational cohort study on 42 patients with Chronic COVID-19 Syndrome (CCS), out of whom 19 also had a chronic fatigue syndrome (ME/CFS). The manuscript describes the clinical characteristics and biomarker findings for these individuals. My first thought based on the Abstract was that a comparison group is needed. The manuscript was previously reviewed by two reviewers who had had the same thought (I have read their comments and the rebuttal).

The choice of the control group depends on the study question. If the main interest is to disentangle the contribution of the SARS-Cov-2 on the outcomes (e.g. laboratory values), then the comparison group should be similar to the study group, but they should be without SARS-Cov-2. The authors hint towards this: "It is unclear yet, if pathomechanisms of post-infectious fatigue syndromes may be different depending on the pathogen." Reviewer #1 suggested choosing a comparison group along these lines: "One important weakness of the study is the lack of an appropriate comparison group in which the same subjective and objective abnormalities were assessed—such as other patients matched for age and gender seeking care at the same time for symptoms unrelated to COVID-19"

If the main interest is to compare the outcomes between COVID-19 patients with and without chronic fatigue, then Reviewer #2's suggestion is more appropriate: "A major weakness of this study is lack of an appropriate control cohort, which ideally would consist of age-matched patients with mild to moderate COVID-19 that did not report chronic fatigue." However, I am not sure this was the purpose of the study. The authors "fully agree" with both Reviewer #1's and Reviewer #2's suggestions. With the data they have, they can only fulfil the request from Reviewer #1 which I think is the more appropriate suggestion for selecting a comparison group in this context.

I think it would be important to include information on the added comparison group in the Abstract as this was a crucial improvement to the manuscript.

I suggest following the guidelines for reporting observational studies and making sure that the items listed on the STROBE checklist (<https://www.strobe-statement.org/checklists/>) are covered in the manuscript.

Were any sample size calculations made prior to the study? What was the reason to restrict the study period when the patients had COVID-19 from March to June 2020?

Please include a reference to your home page, mentioned in the Methods / Study protocol.

Adjustments for multiple comparisons should be performed as appropriate. I suggest removing the current text “A p-value of <0.05 was considered as statistically significant. Due to multiple testing p-values are considered descriptive without adjustments for multiple comparisons.” and adding text about adjustment for multiple comparisons. In case you choose not to adjust the P values, please remove the sentence “A p-value of <0.05 was considered as statistically significant”, remove highlights in bold for p-values <0.05 , and explain in the footnotes that these are unadjusted P values. Please be consistent in presenting P values: add them to all panels in a figure if you present them in one – currently P values are presented for some of the comparisons but not others, e.g. for Figure 1 these are missing from the top two panels in the left column.

Should “We performed the statistical analysis with Excel GraphPad Prism 6.0” read “We performed the statistical analysis with Excel and GraphPad Prism 6.0”?

Point to point response to the referee comments

REVIEWER COMMENTS

Reviewer #1 (Remarks to the Author):

Post-acute COVID-19 syndrome is an emerging and serious health problem, and manuscripts on this topic therefore are welcome.

However, this revised manuscript, transferred from Nature Medicine to Nature Communications, continues to have serious deficiencies. The authors have provided some of the critically important Methods detail that was absent in the first draft of the manuscript. However, as outlined below, four major problems remain.

FIRST PROBLEM. Two reviewers of the previous manuscript cited the importance of having a comparison group of people without COVID-19. The authors state, in their Rebuttal and in the Methods section of this revised manuscript (p. 19), that “A sex and age matched postinfectious non-COVID-19 ME/CFS group with a mean disease onset of 13 months (range 7 - 19 months) was seen and diagnosed during the same period in our outpatient clinic by the same physicians and was selected for comparison.” That is a very appropriate comparison group. However, it would also be valuable to include a HEALTHY comparison group (sex and age matched) that was seronegative for SARS-CoV-2.

Our response:

We agree that a healthy control group would be of interest, too. However, we have not analyzed a healthy control group with these questionnaires and laboratory values yet and this would require some time and funding. We discussed this point under Limitations at the end of the Discussion.

SECOND PROBLEM. While I was delighted to read about the addition of the non-COVID-19 ME/CFS comparison group, I then was very disappointed. The purpose of suggesting a comparison group, obviously, was to compare the putative biomarker results in CCS and CCS/CFS to the results in the comparison group. But where are those results? It is absolutely essential that they be included in Tables 1-3, along with the values for CCS and CCS/CFS. Otherwise, we have little basis for believing that the putative biomarkers really do discriminate the patient groups from other patients who have not suffered from COVID-19.

Our response:

We agree. The patients' characteristics of the non-COVID ME/CFS cohort shown in supplemental Table 2 were added to Table 1 in the revision. In Table 2 and Table 3 the data for the non-COVID ME/CFS cohort was added as well as the data in Figs 1 – 3 previously shown in Suppl. Figures.

THIRD PROBLEM. Tables 2 and 3 show many differences in the results between the CCS and CCS/CFS groups to be statistically significant. However, those tables contain multiple comparisons. Furthermore, when those tables include results from the non-COVID-19 ME/CFS comparison group (see Second Problem), they will include even more comparisons.

In such a situation, the p-value chosen to be “statistically significant” must be adjusted for multiple comparisons. Yet the authors state (p. 21) that “A p-value of <0.05 was considered as statistically significant. Due to multiple testing p-values are considered descriptive without adjustments for multiple comparisons.”

I read this statement to mean that the authors did not use any of the several widely-used techniques to adjust for multiple comparisons. In my opinion, the failure to adjust for multiple comparisons would be indefensible. Without this, the authors cannot claim that ANY of the biomarkers that they say reveal “significant” differences really do reveal such differences. Since such differences are the rationale for submitting this report for publication, this failure is a significant problem.

Our response:

We agree. A correction for multiple testing has now been made in Tables 2 and 3 and mentioned in the Results.

In Material and Methods we added:

Due to multiple testing Kruskal-Wallis test and Dunn's multiple comparisons test were applied for data shown in Table 2 and Benjamini–Hochberg (BH) correction for correlation analyses shown in Table 3, aiming to control a false discovery rate of 5%. Adjusted p -values < 0.05 were considered to provide evidence for a statistically significant result.

FOURTH PROBLEM. The authors examined a number of potential biomarkers to see if they discriminated between the CCS and CCS/CFS groups. Many of these biomarkers are commonly measured and useful in diagnosing disease (e.g., CRP, TSH, vitamin D level). However, others are not: e.g., mannose binding lectin, IL-8 concentration within erythrocytes, serum ACE-1 levels. The authors need to explain WHY these unusual markers were chosen, and what they think abnormalities in these markers say about the underlying biology of CCS.

Our response:

We added information regarding these markers in the section Laboratory parameter in Results, page 10.

There are several smaller issues.

Were the laboratory abnormalities reported on page 9 (for example, thyroid function tests indicating Hashimoto’s thyroiditis) NEW, or had they also been present prior to acute COVID-19? The reader needs to know this.

Our response:

We added this information in the Results: Three of them had been diagnosed with Hashimoto thyroiditis before COVID-19.

The authors should consider changing the name “Chronic COVID-19 Syndrome (CCS)” to “post-acute COVID-19 syndrome (PACS)”, a term that is being used increasingly. This term is preferred because we do not yet know how long the debilitating symptoms following COVID-19 will last: will they last for many years, as with other “chronic” illnesses? If not, the term “chronic” may be misleading. Also, at the first use of the term “post-acute COVID-19 syndrome”, it might be appropriate to add “(also known as ‘long COVID’)”.

Our response:

We agree, but suggest to use the abbreviation PCS for Post COVID Syndrome, which is in accordance with the German and NICE guidelines for long COVID. Patients fulfilling the Canadian Consensus Criteria for CFS are referred to as PCS/CFS.

Reviewer #2 (Remarks to the Author):

In this revised manuscript, the authors address many of the reviewer comments from the prior submission. However, the integration of a control cohort (COVID-19 negative ME/CFS patients) is somewhat incomplete, particularly in terms of figures and tables, and laboratory data is lacking. In addition, there is no addition of a fatigue negative COVID-19 positive cohort, which would allow for a more relevant investigation of disease mechanisms. Presence of similar abnormal laboratory values in a proper control group is a minimum step necessary to suggest a mechanistic connection rather than correlation. The authors suggest that fatigue in CCS/CSF has a different mechanism than fatigue in CCS, but it is unclear in the conclusions whether CCS/CSF should be lumped with COVID-19 negative ME/CFS, or if three different pathophysiological entities are present.

Please see the first answer to Reviewer 1, too.

The patients characteristics of the non-COVID ME/CFS cohort previously shown in supplemental Table 2 were added to Table 1 in the revision as well as the data in Figs 1 – 3 previously shown in Suppl. Figures. In Table 2 and Table 3 the data for the non COVID ME/CFS cohort was added. We comparatively analyzed the laboratory values with hand grip strength in Table 3 for non-COVID ME/CFS, too but had not all laboratory values analyzed in these patients as shown in Table S3 for the PCS cohorts.

We agree that a healthy control group would be of interest, too. However, we have not analyzed a healthy control group with these questionnaires and laboratory values yet and this would require some time and funding. We discussed this point under Limitations at the end of the Discussion.

Reviewer #3 (Remarks to the Author):

This is a prospective observational cohort study on 42 patients with Chronic COVID-19 Syndrome (CCS), out of whom 19 also had a chronic fatigue syndrome (ME/CFS). The manuscript describes the clinical characteristics and biomarker findings for these individuals. My first thought based on the Abstract was that a comparison group is needed. The manuscript was previously reviewed by two reviewers who had had the same thought (I have read their comments and the rebuttal).

The choice of the control group depends on the study question. If the main interest is to disentangle the contribution of the SARS-Cov-2 on the outcomes (e.g. laboratory values), then the comparison group should be similar to the study group, but they should be without SARS-Cov-2. The authors hint towards this: “It is unclear yet, if pathomechanisms of post-infectious fatigue syndromes may be different depending on the pathogen.” Reviewer #1 suggested choosing a comparison group along these lines: “One important weakness of the study is the lack of an appropriate comparison group in which the same subjective and objective abnormalities were assessed—such as other patients matched for age and gender seeking care at the same time for symptoms unrelated to COVID-19”

If the main interest is to compare the outcomes between COVID-19 patients with and without chronic fatigue, then Reviewer #2’s suggestion is more appropriate: “A major weakness of this study is lack of an appropriate control cohort, which ideally would consist of age-matched patients with mild to moderate COVID-19 that did not report chronic fatigue.” However, I am not sure this was the purpose of the study. The authors “fully agree” with both Reviewer #1’s and Reviewer #2’s suggestions. With the data they have, they can only fulfil the request from

Reviewer #1 which I think is the more appropriate suggestion for selecting a comparison group in this context.

I think it would be important to include information on the added comparison group in the Abstract as this was a crucial improvement to the manuscript.

Our response:

This is a good point. We added a sentence in the abstract: Further an age- and sex-matched postinfectious non-COVID-19 myalgic encephalomyelitis/chronic fatigue syndrome (ME/CFS) cohort was comparatively evaluated.

I suggest following the guidelines for reporting observational studies and making sure that the items listed on the STROBE checklist (<https://www.strobe-statement.org/checklists/>) are covered in the manuscript.

Our response:

All items on the STROBE checklist are covered in the revised manuscript.

We therefore adapted the title:

Post COVID-19 Chronic Fatigue Syndrome (ME/CFS) following the first pandemic wave in Germany and biomarkers associated with symptom severity: results from a prospective observational study

We added the following sentence in M&M, page 19:

From a total of 81 patients who contacted the Charité Fatigue Center during this time period for an appointment due to persistent symptoms post COVID-19, 57 were seen in our outpatient clinic and 42 fulfilled the inclusion criteria.

Were any sample size calculations made prior to the study? What was the reason to restrict the study period when the patients had COVID-19 from March to June 2020?

Our response:

We plan to include a total of 160 patients in this prospective observational study. Patients included in this report represent our patients from the 1st wave (in Germany March to June 2020). We added this information in the Introduction section, page 4.

Please include a reference to your home page, mentioned in the Methods / Study protocol.

Our response: We added the reference in the Introduction, page 4: <https://cfc.charite.de/>

Adjustments for multiple comparisons should be performed as appropriate. I suggest removing the current text “A p-value of <0.05 was considered as statistically significant. Due to multiple testing p-values are considered descriptive without adjustments for multiple comparisons.” and adding text about adjustment for multiple comparisons. In case you choose not to adjust the P values, please remove the sentence “A p-value of <0.05 was considered as statistically significant”, remove highlights in bold for p-values <0.05, and explain in the footnotes that these are unadjusted P values. Please be consistent in presenting P values: add them to all panels in a figure if you present them in one – currently P values are presented for some of the comparisons but not others, e.g. for Figure 1 these are missing from the top two panels in the left column.

Our response:

We corrected data shown in Tables 2 and 3 for multiple comparisons.

In Materials&Methods we added: Due to multiple testing Kruskal-Wallis test and Dunn's multiple comparisons test were applied for data shown in Table 2 and Benjamini–Hochberg (BH) correction for correlation analyses shown in Table 3, aiming to control a false discovery rate of 5%. Further we added all p-values in the Figures as suggested.

Should “We performed the statistical analysis with Excel GraphPad Prism 6.0” read “We performed the statistical analysis with GraphPad Prism 6.0”?

Our response:

Yes, thank you, we corrected this.

REVIEWER COMMENTS

Reviewer #1 (Remarks to the Author):

The manuscript is improved, but still contains important problems, in my opinion. [Note: Because the journal's editorial system does not allow underlining, I have used CAPS to emphasize certain words.]

MAJOR PROBLEMS

1) How representative are the 42 PCS and PCS/CFS patients studied, and the ME/CFS patients studied, of larger populations at this institution?

a) The PCS patients were selected from 81 patients responding to a website announcement of a clinic for post-COVID-19-syndrome patients: a convenience sample. However, no systematic follow-up of all COVID-19 patients seen at this institution was conducted, so we can't know how representative the 81 patients were even of those patients cared for at this institution for COVID-19.

b) An inclusion criterion was "absence of relevant cardiac, respiratory, neurological or psychiatric comorbidity", but no systematic evaluation to look for such comorbidity was conducted.

c) We are told that "all patients had to provide proof of previous COVID-19 diagnosis by SARS-CoV-2-PCR... or serology", and then we are told that three did not. (p. 21, lines 496-9).

2) The manuscript has been strengthened by presenting more detailed data from the ME/CFS comparison group, and more clearly distinguishing the PCS and PCS/CFS subgroups. However, the study lacks three other highly relevant comparison groups: 1) healthy, SARS-CoV-2-antibody-negative age/gender-matched controls; 2) age-matched patients with mild to moderate COVID-19 that did not report chronic fatigue; 3) other patients matched for age and gender seeking care at the same time for symptoms unrelated to COVID-19. The absence of these comparison groups substantially limits the conclusions that can be drawn. At a minimum, the Discussion should fully discuss the limitations created by the absence of these comparison groups.

3) The manuscript now compares many different variables between three groups: PCS, PCS/CFS, and ME/CFS. However, this comparison is not presented rigorously:

a) For EVERY variable the result in EACH of the three groups should be reported clearly, in a format (table or figure) that is CONSISTENT across all types of variables, along with the statistical significance of differences between groups. This is not done, making it very difficult for the reader.

b) If there are three groups of patients, then there are THREE comparisons that need to be reported for each variable: PCS vs. PCS/CFS, PCS vs ME/CFS, PCS/CFS vs. ME/CFS. The statistical significance of only TWO of these comparisons is generally provided (PCS vs. PCS/CFS, PCS/CFS vs. ME/CFS), and sometimes

not even this comparison is presented. This should be corrected.

c) The authors report many important variables only in SUPPLEMENTARY tables and figures (e.g., autonomic dysfunction, postural heart rate and BP, all the laboratory values). These data should be presented in the main text tables. The laboratory studies are among the most interesting results: they should not be buried in the supplementary material.

d) Although there is not a healthy comparison group enrolled in this study, for certain variables (e.g., handgrip strength, autonomic dysfunction symptoms, postural heart rate and BP values, certain laboratory values) the authors report that values in the patient groups were “abnormal” compared to previously-established normal values in healthy populations. Citations of the source of these normal values needs to be provided.

MINOR PROBLEMS

1) As summarized above, this paper compares the values of a large number of variables in three different groups: PCS, PCS/CFS, ME/CFS. This also should be done in the Abstract.

For example, the Abstract says “hand grip strength was diminished in most PCS patients” (p. 2, lines 40-1). Diminished compared to what: the ME/CFS comparison group, healthy controls, other? As it stands, the statement is meaningless.

As another example, the Abstract should include the results of the biomarkers in the PCS and ME/CFS groups. These are among the most interesting results.

2) I suggest changing “persistent symptoms following mild COVID-19 referred to as long COVID” (p. 2, lines 51-2) to “persistent symptoms following mild COVID-19, referred to as post-COVID-19 syndrome (PCS) or ‘long COVID’.”

3) I suggest adding “PEM” and “orthostatic intolerance” to the following sentence: “Profound mental and physical fatigue, PEM, cognitive impairment, chronic pain and orthostatic intolerance are key symptoms of ME/CFS” (p.3, lines 70-1). This is important because it explains why the investigators measured both PEM and autonomic function.

4) Table 1 should include the results of statistical comparisons between the three groups, for each variable compared.

5) The text of the Results that discusses the results displayed in Table 1 (pp. 6-7) needs not repeat the numbers in the table, but merely briefly summarize the statistically significant differences.

Indeed, throughout the manuscript, I would urge the authors not to repeat numbers that already appear

in the tables/figures in the text of the Results. This makes the paper longer than it needs to be, and harder to read.

6) Table 3 takes a lot of space, and says relatively little. Since I am suggesting that more data be moved from the supplemental tables and figures to the main tables and figures, the authors should consider making Table 3 a supplemental table.

7) The authors state (p. 15, lines 273-6): “There are simpler diagnostic criteria including the IOM criteria by the Institute of Medicine and the Centers for Disease Control and Prevention (CDC)-1994/Fukuda criteria but they should be used for screening purposes only as both lack key symptoms required by CCC [the Canadian Criteria] for ME/CFS diagnosis.”

I and others would disagree. The authors of the Institute of Medicine (IOM) criteria included many eminent experts in ME/CFS (I was not one of them), and those experts developed the IOM criteria because they felt that the CCC and CDC 1994 criteria had serious limitations. The authors should not imply that most investigators of ME/CFS or PCS regard the CCC as the “gold standard” for criteria.

Reviewer #2 (Remarks to the Author):

The authors addressed many of the reviewer comments from the prior submission including better integration of a COVID-19 negative cohort, which contains some minor differences. No healthy control group was added due to lack of resources. While the potential mechanisms are still unclear, the primary goal of determining whether COVID-19 patients with fatigue meet criteria for chronic fatigue syndrome appears to have been accomplished.

Reviewer #3 (Remarks to the Author):

I thank the authors for responding to my comments. Overall, I am happy with the responses, but I have some further suggestions.

The authors have used the Benjamini-Hochberg (BH) correction to adjust for multiple testing in Table 3 and have added the adjusted p-values (i.e. q-values) in the table as requested. Unadjusted p-values are not of interest and should be removed from the table.

However, in Table 2, the authors only adjust the p-values using Dunn’s test for multiple comparisons, which accounts for the comparison of 3 groups instead of 2. However, it does not account for multiple testing of symptoms (25 in total). Therefore, Table 2 assumes that all tests for the 25 symptoms are independent of each other, which is a strong assumption. I suggest that the authors further adjust the p-

values in Table 2 for the multiplicity of symptoms (using the BH correction).

Similarly, multiple testing corrections should be applied to the Figures as appropriate. It seems that this has not been done.

The use of abbreviations for the different conditions in different parts of the manuscript is somewhat confusing. I think it would be best to remove the abbreviation (ME/CFS) from the title as it is not spelled out in its entirety. In the Abstract, PCS should be spelled out as post COVID-19 syndrome. In the Tables, ME is included in the abbreviation only in the non-COVID condition (ME/CFS) but not with COVID-19 (abbreviated PCS/CFS instead of PCS/ME/CFS). To avoid any confusion, the abbreviations should be used consistently and spelled out in all the Table/Figure footnotes.

We are very grateful to the reviewers for their time and valuable advices, all of which we have implemented. All changes are highlighted by track changes in the manuscript. Additions to tables are highlighted by yellow marking.

REVIEWER COMMENTS

Reviewer #1 (Remarks to the Author):

The manuscript is improved, but still contains important problems, in my opinion. [Note: Because the journal's editorial system does not allow underlining, I have used CAPS to emphasize certain words.]

MAJOR PROBLEMS

1) How representative are the 42 PCS and PCS/CFS patients studied, and the ME/CFS patients studied, of larger populations at this institution?

a) The PCS patients were selected from 81 patients responding to a website announcement of a clinic for post-COVID-19-syndrome patients: a convenience sample. However, no systematic follow-up of all COVID-19 patients seen at this institution was conducted, so we can't know how representative the 81 patients were even of those patients cared for at this institution for COVID-19.

Our response:

We agree that from the design of our study we do not know how representative our subgroup is for Post COVID-19 patients. We will be able to answer this question from an ongoing study in which we analyse 300 randomly selected patients with a positive PCR test in March 2020. A recent study from Italy provides a better estimate of the frequency of ME/CFS in PCS (Mantovani E, 2021). They recruited all PCS patients seen at their clinic <65 years, >6 months follow up after COVID and without comorbidity. From these 37 patients 27% (n=10) fulfilled the International Consensus Criteria for ME/CFS according to Carruthers 2011.

We added these informations in the limitations in the discussion page 15 and added these references: Mantovani E, Mariotto S, Gabbiani D, Dorelli G, Bozzetti S, Federico A, Zanzoni S, Girelli D, Crisafulli E, Ferrari S, Tamburin S. Chronic fatigue syndrome: an emerging sequela in COVID-19 survivors? J Neurovirol. 2021 Aug;27(4):631-637. doi: 10.1007/s13365-021-01002-x. Epub 2021 Aug 2. PMID: 34341960; PMCID: PMC8328351.

From all non-COVID postinfectious ME/CFS patients evaluated during the same period at our clinic (n=123) those with the shortest duration of illness (n=19) were selected.

We added this information in Results, page 6.

b) *An inclusion criterion was “absence of relevant cardiac, respiratory, neurological or psychiatric comorbidity”, but no systematic evaluation to look for such comorbidity was conducted.*

Our response:

We agree that the selection of patients and assessment of inclusion and exclusion criteria was not well described. Part of the evaluation of the patients for inclusion and exclusion criteria was described in the section functional studies which we moved now into the section describing the cohort selection criteria. We revised Materials and Methods page 16 - 18 accordingly.

c) *We are told that “all patients had to provide proof of previous COVID-19 diagnosis by SARS-CoV-2-PCR... or serology”, and then we are told that three did not. (p. 21, lines 496-9).*

Our response:

We had included 3 patients with loss of smell and taste due to the high diagnostic specificity (97%) of this symptom. We added this information in Materials and methods, page 17 and cite the reference: Haehner A, Draf J, Dräger S, de With K, Hummel T. Predictive Value of Sudden Olfactory Loss in the Diagnosis of COVID-19. *ORL J Otorhinolaryngol Relat Spec.* 2020;82(4):175-180. doi: 10.1159/000509143. Epub 2020 Jun 11. PMID: 32526759; PMCID: PMC7360503.

2) *The manuscript has been strengthened by presenting more detailed data from the ME/CFS comparison group, and more clearly distinguishing the PCS and PCS/CFS subgroups. However, the study lacks three other highly relevant comparison groups: 1) healthy, SARS-CoV-2-antibody-negative age/gender-matched controls; 2) age-matched patients with mild to moderate COVID-19 that did not report chronic fatigue; 3) other patients matched for age and gender seeking care at the same time for symptoms unrelated to COVID-19. The absence of these comparison groups substantially limits the conclusions that can be drawn. At a minimum, the Discussion should fully discuss the limitations created by the absence of these comparison groups.*

Our response: We agree that data from further control groups would have been valuable and added this limitation in the discussion, page 15.

3) *The manuscript now compares many different variables between three groups: PCS, PCS/CFS, and ME/CFS. However, this comparison is not presented rigorously: a) For EVERY variable the result in EACH of the three groups should be reported clearly, **in a format (table or figure) that is CONSISTENT across all types of variables, along with the statistical significance of differences between groups.** This is not done, making it very difficult for the reader.*

b) *If there are three groups of patients, then **there are THREE comparisons that need to be reported for each variable: PCS vs. PCS/CFS, PCS vs ME/CFS, PCS/CFS vs. ME/CFS.** The statistical significance of only TWO of these comparisons is generally provided (PCS vs. PCS/CFS, PCS/CFS vs. ME/CFS), and sometimes not even this comparison is presented. This should be corrected.*

Our response: We agree and apologize that the statistical results have not been presented rigorously in the previous version. In the current version, all statistical analyses and presentation of data for all tables and figures are revised with support from Prof. Frank Konietschke, deputy chair of the Charité Institute for Biometry and Clinical Epidemiology, who is added as new coauthor. The statistical methods used were revised and results clearly described. Differences in severity of some symptoms were no longer found after BH correction comparing the 3 cohorts so we omitted the previous Suppl. Fig.5.

c) *The authors report many important variables only in SUPPLEMENTARY tables and figures (e.g., **autonomic dysfunction, postural heart rate and BP, all the laboratory values**). These data should be **presented in the main text tables.** The laboratory studies are among the most interesting results: they should not be buried in the supplementary material.*

Our response: As suggested we present in the revised manuscript in Results the autonomic data as Table 3, postural heart rate and BP as Fig. 4 and the laboratory values as Table 4. In Fig. 4 we added values for 1, 2 and 10 minutes standing and patients with postural tachycardia or hypotension are depicted.

d) *Although there is not a healthy comparison group enrolled in this study, for certain variables (e.g., handgrip strength, autonomic dysfunction symptoms, postural heart rate and BP values, certain laboratory values) the authors report that values in the patient groups were “abnormal” compared to previously-established normal values in healthy populations. Citations of the source of these normal values needs to be provided.*

Our response: These references are cited: ref 23, 24, 25, and 26 in Results and Material & Methods. Normal reference values for laboratory parameter are shown in Table 4.

MINOR PROBLEMS

1) *As summarized above, this paper compares the values of a large number of variables in three different groups: PCS, PCS/CFS, ME/CFS. This also should be done in the Abstract.*

For example, the Abstract says “hand grip strength was diminished in most PCS patients” (p. 2, lines 40-1). Diminished compared to what: the ME/CFS comparison group, healthy controls, other? As it

stands, the statement is meaningless. As another example, the Abstract should include the results of the biomarkers in the PCS and ME/CFS groups. These are among the most interesting results.

Our response: We added these information in the Abstract.

2) I suggest changing “persistent symptoms following mild COVID-19 referred to as long COVID” (p. 2, lines 51-2) to “persistent symptoms following mild COVID-19, referred to as post-COVID-19 syndrome (PCS) or ‘long COVID’.”

Our response: Thanks for this advice, which we changed accordingly on page 2.

3) I suggest adding “PEM” and “orthostatic intolerance” to the following sentence: “Profound mental and physical fatigue, PEM, cognitive impairment, chronic pain and orthostatic intolerance are key symptoms of ME/CFS” (p.3, lines 70-1). This is important because it explains why the investigators measured both PEM and autonomic function.

Our response: Thanks for this advice, which we changed accordingly on page 3.

*4) **Table 1 should include the results of statistical comparisons between the three groups, for each variable compared.***

Our response: Thank you for the remark. We added and adapted the statistical comparisons, as suggested. Following BH correction all p-values are =1. We added this information in the Table. Bell and CFQ scores are now shown in Fig. 1 only.

5) The text of the Results that discusses the results displayed in Table 1 (pp. 6-7) needs not repeat the numbers in the table, but merely briefly summarize the statistically significant differences. Indeed, throughout the manuscript, I would urge the authors not to repeat numbers that already appear in the tables/figures in the text of the Results. This makes the paper longer than it needs to be, and harder to read.

Our response: As suggested we deleted numbers in Results.

*6) **Table 3 takes a lot of space, and says relatively little.** Since I am suggesting that more data be moved from the supplemental tables and figures to the main tables and figures, the authors should consider making Table 3 a supplemental table.*

Our response: As suggested we revised the presentation of the data of the previous Table 3 (correlations HGS with laboratory parameters). As suggested by our statistician Frank Konietschke we show the data now as correlation plots, which makes their presentation much clearer and simpler. We therefore suggest presenting this data as new Fig. 5 in the main text. Alternatively, it could be put into

the supplement. The correlation of fatigue questionnaires with thrombocytes and thyroid tests did not remain significant after BH correction, therefore we decided not to mention this negative data in the revision and omitted previous Suppl. Table 4 and the respective section in the discussion.

7) The authors state (p. 15, lines 273-6): "There are simpler diagnostic criteria including the IOM criteria by the Institute of Medicine and the Centers for Disease Control and Prevention (CDC)-1994/Fukuda criteria but they should be used for screening purposes only as both lack key symptoms required by CCC [the Canadian Criteria] for ME/CFS diagnosis."

I and others would disagree. The authors of the Institute of Medicine (IOM) criteria included many eminent experts in ME/CFS (I was not one of them), and those experts developed the IOM criteria because they felt that the CCC and CDC 1994 criteria had serious limitations. The authors should not imply that most investigators of ME/CFS or PCS regard the CCC as the "gold standard" for criteria.

Our response: As suggested we changed this section in Discussion, page 12/13.

Reviewer #2 (Remarks to the Author):

The authors addressed many of the reviewer comments from the prior submission including better integration of a COVID-19 negative cohort, which contains some minor differences. No healthy control group was added due to lack of resources. While the potential mechanisms are still unclear, the primary goal of determining whether COVID-19 patients with fatigue meet criteria for chronic fatigue syndrome appears to have been accomplished.

Our response: No further changes are required.

Reviewer #3 (Remarks to the Author):

I thank the authors for responding to my comments. Overall, I am happy with the responses, but I have some further suggestions.

The authors have used the Benjamini-Hochberg (BH) correction to adjust for multiple testing in Table 3 and have added the adjusted p-values (i.e. q-values) in the table as requested. Unadjusted p-values are not of interest and should be removed from the table.

Our response: As suggested we removed the unadjusted p-values. Further we revised the presentation of the data of the previous Table 3 (correlations HGS with laboratory parameters) as new Fig. 5. As suggested by our statistician we show the data now as correlation plots, which makes their presentation much clearer and simpler.

However, in **Table 2**, the authors only adjust the p-values using Dunn's test for multiple comparisons, which accounts for the comparison of 3 groups instead of 2. However, it does not account for multiple testing of symptoms (25 in total). Therefore, Table 2 assumes that all tests for the 25 symptoms are independent of each other, which is a strong assumption. I suggest that the authors further adjust the p-values in Table 2 for the multiplicity of symptoms (using the BH correction). Similarly, multiple testing corrections should be applied to the Figures as appropriate. It seems that this has not been done.

Our response: As suggested we performed BH corrections. Please see also our response to reviewer 1 major point 3 and minor point 6 regarding the statistics and new Fig. 5.

The use of abbreviations for the different conditions in different parts of the manuscript is somewhat confusing. I think it would be best to remove the abbreviation (ME/CFS) from the title as it is not spelled out in its entirety. In the Abstract, PCS should be spelled out as post COVID-19 syndrome. In the Tables, ME is included in the abbreviation only in the non-COVID condition (ME/CFS) but not with COVID-19 (abbreviated PCS/CFS instead of PCS/ME/CFS). To avoid any confusion, the abbreviations should be used consistently and spelled out in all the Table/Figure footnotes.

Our response: As suggested we skip ME/CFS in the title and use the abbreviation PCS/non-ME/CFS and PCS/ME/CFS throughout the manuscript. In the Abstract we spell out post COVID syndrome but suggest to keep the abbreviation PCS, which we use in the abbreviations PCS/nonME/CFS and PCS/ME/CFS.

REVIEWER COMMENTS

Reviewer #4 (Remarks to the Author):

- What are the noteworthy results?

This is a descriptive account of patients with symptoms after COVID who have been divided into a group with a diagnosis of ME/CFS compared to those who do not meet the Canadian Criteria for ME/CFS and then compared to patients with ME/CFS. It is a small study and there are issues with the methodology.

- Will the work be of significance to the field and related fields? How does it compare to the established literature? If the work is not original, please provide relevant references.

The authors state [ref 44]: A recent study from Italy provides a better estimate of the frequency of ME/CFS in PCS.

- Does the work support the conclusions and claims, or is additional evidence needed?

Some of the conclusions and claims should be toned down. For example, the association of hand grip strength with laboratory markers. There is no sample size calculation and multiple tests were conducted. It is unclear if this is false positive.

In addition, care needs to be taken in the discussion as in some areas, the authors appear to be quoting potential mechanisms as established fact, when the references used appear to be opinion pieces. For example, the authors say (Discussion p15, line 292 “there is evidence for autoimmunity triggered by COVID19” Reference 35. This reference is an opinion piece and does not appear to suggest that Post COVID syndrome is caused by auto-immunity.

There are a few circular arguments which have not been addressed. So, those fulfilling the Canadian criteria for ME/CFS are likely to have orthostatic symptoms as these are one of a group of symptoms required to have a diagnosis. These are then compared to those fulfilling or not filling the criteria. And the difference in PEM between the ME/CFS group and the non ME/CFS group is not particularly surprising given it is part of the definition.

- Are there any flaws in the data analysis, interpretation and conclusions? - Do these prohibit publication or require revision?

As the authors allude to in the discussion, it seems likely that any differences are going to be the longer disease duration of 13 months rather than differences in pathophysiology. This needs to be discussed in more detail. Disease duration could have been used to adjust for symptom prevalence between the groups.

- Is the methodology sound? Does the work meet the expected standards in your field?

I have some concerns about the methodology:

1. I could not find evidence that the authors had conducted a sample size calculation before conducting these analyses. This is particularly important given the number of analyses conducted. It is not clear to me if the findings of the association between hand grip strength and laboratory findings is real or

chance. I suspect this study is underpowered to show differences between the groups

2. The authors say that all patients required proof of previous infection (confirmed diagnosis of mild to moderate COVID-19), however 3 patients did not require proof because they had loss of smell and taste.

I am concerned that the inclusion criteria are unclear and there may be selection bias. This was mentioned by reviewer 1 and whilst the authors have tried to address it by arguing that it is a sensitive symptom, I have concerns about recruitment.

3. It wasn't clear to me if patients who had abnormal lung function because of COVID infection were then excluded (page 18). If so, this would mean that those with lung damage because of COVID were excluded.

4. I do not believe the Bell disability scale has been validated.

5. I do not believe the measure used to assess post exertional malaise has been validated. Cotler et al also report sensitivity of 81% compared to MS and polio so the authors should at least comment on whether the sensitivity is sufficient to compare between two similar groups of a much smaller sample size than Cotler used.

Reviewer #5 (Remarks to the Author):

This manuscript by Kedor et al. investigated COVID-19 and persistent fatigue syndrome fulfilling the Canadian Consensus Criteria (CCC) of myalgic encephalomyelitis/chronic fatigue syndrome ME/CFS along its specific characteristics. This investigation was carried out among young individuals following mild to moderate COVID-19 infectious disease. In this study, mental health was not a predominant condition as only a minority of patients showed evidence for depression or anxiety.

Authors report on a total of 42 patients who presented to the Charité Fatigue Center with PCS all suffering from persistent moderate to severe fatigue and exertion intolerance six months after diagnosis of COVID-19. Most patients had mild COVID-19 (n=32) and ten had moderate COVID-19 due to pneumonia (per WHO criteria). The study needs some clarifications that I outline below.

The study design needs to be described as an observational study design using a convenient patients study sample. This design needs to be transparent to readers.

The study population description in the main text needs to include the calendar time frame of data collection as Sars-Cov-2 variants may likely have different impact. I suggest that authors include a consort like diagram of their study population with inclusion and exclusion.

Table 1 needs to footnote the source of the patients including the non-COVID ME-CFS and calendar time frame so that it stands on its own.

Selection bias. Reviewer 1 has brought up in his/her comment #2 the issue of using appropriate controls. I would stress further the need for proper sampling strategies that could have mitigated selection bias. Because any factors that influence sample selection themselves influence the variables of interest, the relationship between these variables of interest can become misleading (1). I suggest that authors limit their comparisons to clinical factors that do not have a role in reverse causation when possible. If

authors deem that presenting all factors is necessary, I suggest that author acknowledge explicitly the potential of collider bias (1).

Moreover, as in all observational studies, there could be unknown confounders that could have biased the associations.

Reviewer 3 comments were properly addressed. I suggest that authors consider including only statistically significant comparisons with asterisks - the Benjamini-Hochberg (B-H) statistically significant ones as: $\alpha = 0.05$; * < 0.05 ; ** < 0.001 ; *** < 0.0001

Furthermore, it is worth noting that the BH method assumes independent test and it is likely that these symptoms are correlated and explain the same underlying phenomenon, thus the correction is conservative. Although this is post-hoc at this late stage, data dimension reduction techniques (clustering analysis of factors or redundancy analysis) would have been helpful to minimize the number of tests without using univariate analysis (univariate meaning using the outcome ME/CFS status comparisons which is not recommended as it would be univariate filtering).

References: 1. Griffith GL et al. Collider bias undermines our understanding of COVID-19 disease risk and severity. Nat Commun. 2020 Nov 12;11(1):5749.doi: 10.1038/s41467-020-19478-2.

REVIEWER COMMENTS

Reviewer #4 (Remarks to the Author):

- *What are the noteworthy results?*

This is a descriptive account of patients with symptoms after COVID who have been divided into a group with a diagnosis of ME/CFS compared to those who do not meet the Canadian Criteria for ME/CFS and then compared to patients with ME/CFS. It is a small study and there are issues with the methodology.

- *Will the work be of significance to the field and related fields? How does it compare to the established literature? If the work is not original, please provide relevant references.*

The authors state [ref 44]: A recent study from Italy provides a better estimate of the frequency of ME/CFS in PCS.

- *Does the work support the conclusions and claims, or is additional evidence needed?*

Some of the conclusions and claims should be toned down. For example, the association of hand grip strength with laboratory markers. There is no sample size calculation and multiple tests were conducted. It is unclear if this is false positive.

In addition, care needs to be taken in the discussion as in some areas, the authors appear to be quoting potential mechanisms as established fact, when the references used appear to be opinion pieces. For example, the authors say (Discussion p15, line 292 “there is evidence for autoimmunity triggered by COVID19” Reference 35. This reference is an opinion piece and does not appear to suggest that Post COVID syndrome is caused by auto-immunity.

There are a few circular arguments which have not been addressed. So, those fulfilling the Canadian criteria for ME/CFS are likely to have orthostatic symptoms as these are one of a group of symptoms required to have a diagnosis. These are then compared to those fulfilling or not filling the criteria. And the difference in PEM between the ME/CFS group and the non ME/CFS group is not particularly surprising given it is part of the definition.

Our response:

We thank for these notes:

To tone down the conclusion we added in the Abstract and on page 15 and 16 “may” in the following sentences:

“Association of HGS with hemoglobin (Hb), IL-8 and CRP in PCS/non-ME/CFS and with Hb, NT-pro BNP, bilirubin, and ferritin in PCS/ME/CFS “may” indicate low level inflammation and hypoperfusion as potential pathomechanisms.”

“Consistently with our clinical observations these associations “may” point to common and distinct pathomechanisms in PCS/non-ME/CFS and PCS/ME/CFS patients.”

“In the PCS/non-ME/CFS cohort, a negative correlation of Fmax1 with IL-8 in erythrocytes and CRP levels with Fmean2 was found which “may” point to low level inflammation as mechanism of muscle fatigue.”

In the Discussion, page 17 we changed “point to” into “which may indicate” in this sentence:

“By defining and characterizing subgroups of PCS patients we could identify associations of HGS with biomarkers “which may indicate” hypoperfusion and inflammation as potential pathomechanisms.”

We exchanged Ref. 35, now reference #36 for a study analysing autoantibodies by an antigen array chip:

Rojas M, Rodríguez Y, Acosta-Ampudia Y, Monsalve DM, Zhu C, Li QZ, Ramírez-Santana C, Anaya JM. Autoimmunity is a hallmark of post-COVID syndrome. J Transl Med. 2022 Mar 16;20(1):129. doi: 10.1186/s12967-022-03328-4. PMID: 35296346; PMCID: PMC8924736.

To avoid circular arguments we added the following sentences or sentence additions (underlined in tracked changes version):

Page 7: “As expected due to the diagnostic criteria frequency and severity of PEM as the cardinal symptom of ME/CFS was a strong discriminatory factor between PCS/non-ME/CFS and non-COVID ME/CFS patients, but differences were not significant between PCS/non-ME/CFS and PCS/ME/CFS.”

Page 13: “Postural tachycardia and hypotension was noted more frequent in PCS/ME/CFS, which was not unexpected as orthostatic symptoms are a hallmark of ME/CFS.”

- Are there any flaws in the data analysis, interpretation and conclusions? - Do these prohibit publication or require revision?

As the authors allude to in the discussion, it seems likely that any differences are going to be the longer disease duration of 13 months rather than differences in pathophysiology. This needs to be discussed in more detail. Disease duration could have been used to adjust for symptom prevalence between the groups.

Our response:

We thank for this note. It was not originally planned to add a non-COVID ME/CFS patient cohort, but was suggested from Reviewer #1. As ME/CFS patients are usually referred to us later in the disease course, the group we had seen during the same time had a longer disease duration from 7 – 19 months in contrast to 6 months in the PCS cohort. Therefore we did not adjust for disease duration.

We discuss this limitation on page 16 and added a further sentence on page 13 (underlined in tracked changes version):

“A possible confounder might be the longer disease duration in the latter group.”

- Is the methodology sound? Does the work meet the expected standards in your field?

I have some concerns about the methodology:

1. I could not find evidence that the authors had conducted a sample size calculation before conducting these analyses. This is particularly important given the number of analyses conducted. It is not clear to me if the findings of the association between hand grip strength and laboratory findings is real or chance. I suspect this study is underpowered to show differences between the groups

Our response:

Many thanks for the valuable comment. Since the study is observational and not confirmatory a formal sample size calculation by a power analysis is not possible. In this study we aim to characterize potential effects and factors and not to confirm any. As we mentioned above Reviewer #1 suggested to add a further cohort of non-COVID ME/CFS. Several differences found in the original paper did not reach statistical significance by performing multiple correction including this 3rd group. However, our findings clearly indicate potential effects that would require a confirmatory study powered based on the findings in current study.

We agree that the study comparing the 3 cohorts is underpowered to show differences. We therefore suggest to add from the original manuscript as a supplement Figure S6 comparing symptom severity of the two PCS cohorts only showing significant differences for stress intolerance and hypersensitivity to noise, light and temperature, which is no longer significant when comparing the 3 groups. We added this information in the results, page 6 and discussion, page 13:

“When comparing the two PCS cohorts only the higher symptom burden for stress intolerance, and hypersensitivity to temperature, noise and light in the ME/CFS cohort was significant, too (see Suppl Figure S6).”

2. The authors say that all patients required proof of previous infection (confirmed diagnosis of mild to moderate COVID-19), however 3 patients did not require proof because they had loss of smell and taste. I am concerned that the inclusion criteria are unclear and there may be selection bias. This was mentioned by reviewer 1 and whilst the authors have tried to address it by arguing that it is a sensitive symptom, I have concerns about recruitment.

Our response:

As PCR tests were not available for everyone in the beginning of pandemic we had as further inclusion criteria symptoms of loss of smell and/or taste.

We added this information on page 18: “3 patients were included with typical initial symptoms of loss of smell and taste due to the high diagnostic specificity of this symptom in accordance with the inclusion criteria of our study protocol.”

3. It wasn't clear to me if patients who had abnormal lung function because of COVID infection were then excluded (page 18). If so, this would mean that those with lung damage because of COVID were excluded.

Our response: We added in the methodology part, page 18: “Patients with impaired diffusion capacity were not included in the study.”

4. I do not believe the Bell disability scale has been validated.

Our response: The Bell disability scale is a self-scoring symptom scale and was used descriptive.

5. I do not believe the measure used to assess post exertional malaise has been validated. Cotler et al also report sensitivity of 81% compared to MS and polio so the authors should at

least comment on whether the sensitivity is sufficient to compare between two similar groups of a much smaller sample size than Cotler used.

Our response: The PEM score is not validated and was used descriptively.

Reviewer #5 (Remarks to the Author):

This manuscript by Kedor et al. investigated COVID-19 and persistent fatigue syndrome fulfilling the Canadian Consensus Criteria (CCC) of myalgic encephalomyelitis/chronic fatigue syndrome ME/CFS along its specific characteristics. This investigation was carried out among young individuals following mild to moderate COVID-19 infectious disease. In this study, mental health was not a predominant condition as only a minority of patients showed evidence for depression or anxiety.

Authors report on a total of 42 patients who presented to the Charité Fatigue Center with PCS all suffering from persistent moderate to severe fatigue and exertion intolerance six months after diagnosis of COVID-19. Most patients had mild COVID-19 (n=32) and ten had moderate COVID-19 due to pneumonia (per WHO criteria). The study needs some clarifications that I outline below.

The study design needs to be described as an observational study design using a convenient patients study sample. This design needs to be transparent to readers.

The study population description in the main text needs to include the calendar time frame of data collection as Sars-Cov-2 variants may likely have different impact. I suggest that authors include a consort like diagram of their study population with inclusion and exclusion.

Table 1 needs to footnote the source of the patients including the non-COVID ME-CFS and calendar time frame so that it stands on its own.

Our response:

Our study is a substudy of the Pa-COVID-19 study, a prospective observational cohort study assessing pathophysiology and clinical characteristics of patients with COVID-19 at Charité Universitätsmedizin Berlin. We added this information in the introduction and the reference #19 for this study (Kurth et al., 2020). Regarding the patients study sample we refer to the response to R#4 at the end of page 3 of this document.

We added the following sentences in the introduction, page 4:

“Our study is a substudy of the Pa-COVID-19 study, a prospective observational cohort study assessing pathophysiology and clinical characteristics of patients with COVID-19 at Charité Universitätsmedizin Berlin (Kurth et al., 2020).”

“Patients had COVID between March and June 2020 when there were no variants of SARS-Cov2 reported in our region.”

We added a footnote as suggested to Table 1.

In the methods part, page 18 it is stated: “All patients enrolled in this study presented at our outpatient clinics between August 2020 and November 2020.”

Further we added a Consort flow diagram as supplemental Fig S7.

Selection bias. Reviewer 1 has brought up in his/her comment #2 the issue of using appropriate controls. I would stress further the need for proper sampling strategies that could have mitigated selection bias. Because any factors that influence sample selection themselves influence the variables of interest, the relationship between these variables of interest can become misleading (1). I suggest that authors limit their comparisons to clinical factors that do not have a role in reverse causation when possible. If authors deem that presenting all factors is necessary, I suggest that author acknowledge explicitly the potential of collider bias (1). Moreover, as in all observational studies, there could be unknown confounders that could have biased the associations.

Reference: Griffith GL et al. Collider bias undermines our understanding of COVID-19 disease risk and severity. Nat Commun. 2020 Nov 12;11(1):5749.doi: 10.1038/s41467-020-19478-2.

Our response:

We thank for this advice and agree that our study has a selection bias due to the inclusion criteria and therefore a potential bias due to uncontrolled confounders. We thank the reviewer for providing the reference and added the following sentence as a limitation on page 17: “By nature of the study, the results might be biased due to uncontrolled confounders (Ref. # 47 Griffith GL et al, 2020).”

*Reviewer 3 comments were properly addressed. I suggest that authors consider including only statistically significant comparisons with asterisks - the Benjamini-Hochberg (B-H) statistically significant ones as: =0.05 ; * <0.05 ; ** <0.001; ***<0.0001*

Furthermore, it is worth noting that the BH method assumes independent test and it is likely that these symptoms are correlated and explain the same underlying phenomenon, thus the correction is conservative. Although this is post-hoc at this late stage, data dimension reduction techniques (clustering analysis of factors or redundancy analysis) would have been helpful to minimize the number of tests without using univariate analysis (univariate meaning using the outcome ME/CFS status comparisons which is not recommended as it would be univariate filtering).

Our response:

Many thanks for the comments that we completely agree with. We used the BH correction upon request from previous review rounds. We are aware of its assumptions and discussed it critically in our group. However, we used the correction in an exploratory manner only (due to the large amount of comparisons made). During the data analysis time we indeed performed methods for dimension reduction by screening. The used maximum tests can be used in that manner as a filtering method by first filtering. Here, all dimensions are excluded that do not exceed a certain bound. However, since the sample sizes are so small these methods may be unstable and especially hard to interpret. We furthermore also applied the

Benjamini-Yekutieli correction, which is valid under general dependence. The method is too conservative and estimated all p-values = 1. We therefore decided to run the analysis as presented. We added a critical remark about our analysis to the discussion section.

We adapted Table 2 as suggested with asterisk indicating significant changes and n.s. for those not significant. In Table 3 we omitted the p-values as none remained significant and added this information in the footnote.

REVIEWERS' COMMENTS

Reviewer #5 (Remarks to the Author):

I have no further additional queries. The manuscript has substantially improved. The authors did a thorough and good job in responding to reviewers' comments.

REVIEWERS' COMMENTS

Reviewer #5 (Remarks to the Author):

I have no further additional queries. The manuscript has substantially improved. The authors did a thorough and good job in responding to reviewers' comments.

Many thanks for your response.

The final corrections made are listed on the authors` checklist.

In summary,

We adapted the title and abstract to your guidance, we updated affiliations, we rearranged the order of sections, we added n in all figures and updated legends, we added reference numbers and page numbers to supplemental material

Yours sincerely

Claudia Kedor

on behalf of all authors